# ParetoQ: Improving Scaling Laws in Extremely Low-bit LLM Quantization

Zechun Liu[1] [†]    Changsheng Zhao[1]    Hanxian Huang[1]    Sijia Chen[1]    Jing Zhang[1]
Jiawei Zhao[1]    Scott Roy[1]    Lisa Jin[1]    Yunyang Xiong[1]    Yangyang Shi[1]    Lin Xiao[1]
Yuandong Tian[1]    Bilge Soran[1]    Raghuraman Krishnamoorthi[1]    Tijmen Blankevoort[1]
Vikas Chandra[1]

[1]Meta AI

## Abstract

The optimal bit-width for achieving the best trade-off between quantized model size and accuracy has been a subject of ongoing debate. While some advocate for 4-bit quantization, others propose that 1.58-bit offers superior results. However, the lack of a cohesive framework for different bits has left such conclusions relatively tenuous. We present `ParetoQ`, the first unified framework that facilitates rigorous comparisons across 1-bit, 1.58-bit, 2-bit, 3-bit, and 4-bit quantization settings. Our findings reveal a notable learning transition between 2 and 3 bits: *For 3-bits and above, the fine-tuned models stay close to their original pre-trained distributions, whereas for learning 2-bit networks or below, the representations change drastically.* By optimizing training schemes and refining quantization functions, `ParetoQ` surpasses all previous methods tailored to specific bit widths. Remarkably, our `ParetoQ` ternary 600M-parameter model even outperforms the previous SoTA ternary 3B-parameter model in accuracy, using only one-fifth of the parameters. Extensive experimentation shows that ternary, 2-bit, and 3-bit quantization maintains comparable performance in the size-accuracy trade-off and generally exceeds 4-bit and binary quantization. Considering hardware constraints, 2-bit quantization offers promising potential for memory reduction and speedup.

## 1 Introduction

As deep learning continues to scale toward larger models and datasets, significant attention has been devoted to studying the scaling laws that trade-off between model and dataset size to optimize performance and computational efficiency [Hoffmann et al., 2022, Kumar et al., 2024, Dettmers and Zettlemoyer, 2023]. In the meantime, the field is shifting toward lower-precision computation, particularly in large language models, driven by the substantial benefits of memory savings and computational efficiency [Liu et al., 2023a, Ma et al., 2024]. This shift necessitates a rethinking of scaling laws to account for the effects of quantization on resulting quantized model performance.

When allowing for lower-bit quantization, we can freely trade off the bit-width and the number of parameters. Keeping the amount of memory used the same, we could have an 8-bit model, or a 4-bit model twice the size. This begs the question: *What is the optimal trade-off between bit-width and model size?* Recent papers [Dettmers and Zettlemoyer, 2023, Kumar et al., 2024] on scaling laws for low-precision conclude that 4 or 6-bit quantization often resides on the Pareto frontier to balance accuracy and efficiency. Other studies [Ma et al., 2024, Kaushal et al., 2024] suggest that bit-widths as low as 1.58-bit per parameter hold significant promise for the optimal scaling law trade-off. These opposing conclusions highlight the challenges of studying scaling laws in the low-precision domain.

---

[†]Correspondence to: Zechun Liu zechunliu@meta.com .

39th Conference on Neural Information Processing Systems (NeurIPS 2025).

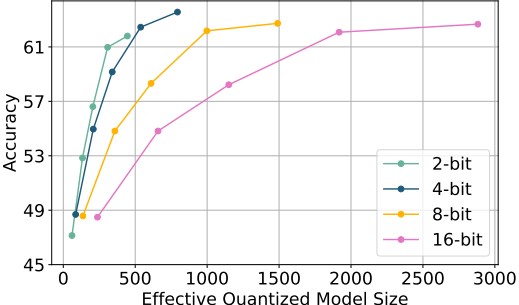
Figure 1: Pareto curves of accuracy-size trade-offs.

In this paper, we demonstrate that previous conclusions on the low-bit scaling laws can be significantly sharpened by better quantization scheme design and training improvements. While previous works define the search space of the QAT scaling laws solely as a function of model parameters ($\mathcal{N}$), token count ($\mathcal{D}$), and quantization precision ($\mathcal{P}$), **we emphasize the critical role that the training scheme ($\mathcal{S}_{\text{train}}$) and the bit-specific quantization function ($\mathcal{F}$) play in the equation**. We formalize the search space as $\mathcal{L}(\mathcal{N}, \mathcal{D}, \mathcal{P}, \mathcal{S}_{\text{train}}, \mathcal{F})$, comprising five dimensions.

To disentangle these complexities, we first identify the optimal training strategy for plausible quantization functions in each bit width, $\mathcal{L}(N, \mathcal{D}, \mathcal{S}_{\text{train}} \mid \mathcal{P}, \mathcal{F})$. Subsequently, with the optimal training strategy ($\mathcal{S}_{\text{train}}^*$) and the token count ($\mathcal{D}^*$) required for saturation, we determine the best quantization function for each bit, $\mathcal{L}(N, \mathcal{F} \mid \mathcal{P}, \mathcal{D}^*, \mathcal{S}_{\text{train}}^*)$. Results highlight that **quantization grids and ranges are pivotal in the sub-4-bit regime, with a sharp learning behavior transition between 1-bit/1.58-bit/2-bit and 3-bit/4-bit.**

Based on the findings, we derive `ParetoQ`, the first framework that unifies the training and quantization scheme in sub 4-bit regime. Rather than fitting hypothetical scaling laws for quantization, **`ParetoQ` demonstrate its robustness by yielding state-of-the-art (SOTA) models at all bit widths**, surpassing prior works tailored for individual bit levels.

These SOTA points in the Pareto chart ensure that our scaling law comparisons are both reliable and consistent, as they derive from homogeneous settings. Leveraging `ParetoQ`, we identify the optimal bit-width for minimizing loss within the effective quantized model size, $\mathcal{L}(\mathcal{N}, \mathcal{P} \mid \mathcal{F}^*, \mathcal{D}^*, \mathcal{S}_{\text{train}}^*)$. Our scaling laws reveal that binary quantization significantly compromises accuracy, while ternary, 2-bit and 3-bit quantization are tied in performance, often surpassing 4-bit. The tiebreaker lies in the kernel implementation, which drives real memory savings and speedups. 1.58-bit and 3-bit quantization are in general less hardware-friendly than 2-bit. We implemented an optimized 2-bit CPU kernel and our results indicate that 2-bit quantization achieves higher speed at the same accuracy compared to 4-bit.

The key contributions of this study are as follows:

• We present a comprehensive study on the intertwined effects of QAT budget allocation and specific choices of quantization functions across 8 models (125M to 3B) and 5 quantization strategies. Our study highlights the unique characteristics and challenges of binary, ternary, and 2/3/4-bit quantization, offering actionable insights and best practices for achieving optimal accuracy-efficiency trade-offs.

• We introduce `ParetoQ`, the first systematic, apples-to-apples comparison of quantization functions at extreme low-bit settings. Each point in the Pareto chart outperforms prior methods optimized for specific bit widths. Specifically, the 1.58-bit `ParetoQ` LLaMA-3 8B model reduces the performance gap to full precision by relatively 37.8% compared to the 1-bit Era's LLaMA-3 8B model [Ma et al., 2024], while using only 30% of the training tokens.

• Our research highlights the potential of 2-bit quantization as a prospective alternative to the traditional 4-bit approach, offering improved accuracy-size trade-off, as underlined in Figure 1. Preliminary speed benchmarks also demonstrate promising efficiency gains with 2-bit quantization. Nevertheless, widespread adoption will require community-wide efforts, such as INT2 support in NVIDIA tensor cores, to unlock the full benefits of 2-bit quantization.

## 2 A Better QAT Scheduling Strategy for Extreme Low-Bit LLMs

In this work, we systematically investigate trade-offs involving bit precision ($\mathcal{P}$), quantization functions ($\mathcal{F}$), model size ($\mathcal{N}$), training strategies ($\mathcal{S}_{train}$) and training token ($\mathcal{D}$): $\mathcal{L}(\mathcal{P}, \mathcal{F}, \mathcal{N}, \mathcal{S}_{train}, \mathcal{D})$.

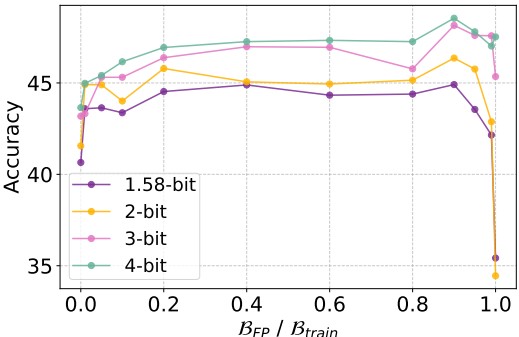

Figure 2: With a fixed total training budget of 100B tokens ($\mathcal{B}_{\text{train}}$), where $\mathcal{B}_{\text{FP}} + \mathcal{B}_{\text{QAT}} = \mathcal{B}_{\text{train}}$, we explore optimal allocation between full-precision pretraining ($\mathcal{B}_{\text{FP}}$) and QAT fine-tuning ($\mathcal{B}_{\text{QAT}}$). "0.0" represents QAT from scratch, while "1.0" indicates full-precision pretraining followed by PTQ. Results on MobileLLM-125M show peak accuracy with ∼90% of the budget for full-precision pretraining and ∼10% for QAT fine-tuning.

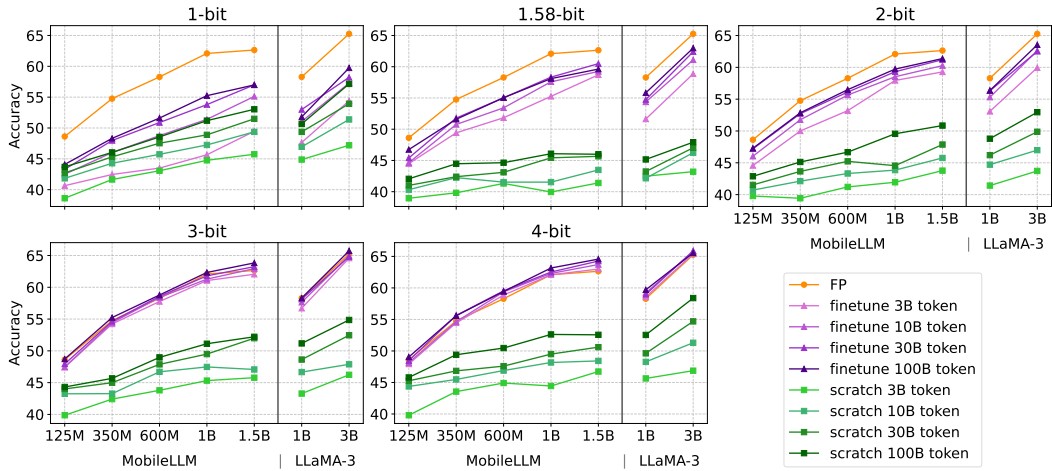

Figure 3: Analysis of training token requirements for quantization-aware fine-tuning and training from scratch across 1-bit, 1.58-bit, 2-bit, 3-bit, and 4-bit settings. Fine-tuning typically saturates at 10B tokens for 3-bit and 4-bit, and at 30B tokens for 1-bit, 1.58-bit, and 2-bit. Fine-tuning consistently outperforms training from scratch in both accuracy and token efficiency across all bit configurations.

Given the vast search space defined by these variables, we first fix the quantization method ($\mathcal{F}$) and explore the dimensions of bit precision ($\mathcal{P}$), training strategies ($\mathcal{S}_{train}$) and training tokens ($\mathcal{D}$) in this section.

## 2.1 Training Budget Allocation

Post-Training Quantization (PTQ) and Quantization-Aware Training (QAT) are two primary quantization approaches. PTQ applies quantization after full-precision training, simplifying deployment but often leads to significant performance loss at bit widths below 4 bits. In contrast, QAT incorporates quantization during training to optimize model performance for low-bit-width representations.

Here we start by answering a key question:

**Given a fixed training budget (in #tokens) $\mathcal{B}_{\textbf{train}} = \mathcal{B}_{\textbf{FPT}} + \mathcal{B}_{\textbf{QAT}}$, how should the budget be optimally allocated between full-precision training ($\mathcal{B}_{\textbf{FPT}}$) and quantization-aware training/finetuning ($\mathcal{B}_{\textbf{QAT}}$) to maximize the accuracy of the quantized model?**

This question is both technically intriguing and practically significant. Our approach begins with analyzing the pretraining phase to determine the optimal switching point from FPT to QAT, aiming to minimize the loss:

$$\mathcal{B}^*_{\text{FPT}}, \mathcal{B}^*_{\text{QAT}} = \underset{\mathcal{B}_{\text{FPT}}+\mathcal{B}_{\text{QAT}}=\mathcal{B}_{\text{train}}}{\arg\min} \mathcal{L}(\mathcal{B}_{\text{FPT}}, \mathcal{B}_{\text{QAT}}|\mathcal{N}, \mathcal{P}) \qquad (1)$$

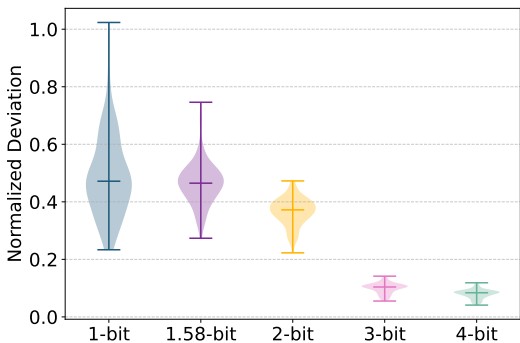

Figure 4: L1 norm difference between QAT-finetuned weights and full-precision initialization ($||W_{\text{finetune}} - W_{\text{init}}||_{l1}$ / $||W_{\text{init}}||_{l1}$). Models quantized to 1, 1.58, and 2 bits show larger weight changes, indicating distinct '*compensation*' behavior in higher-bit quantization and '*reconstruction*' in lower-bit settings.

where $\mathcal{B}_{\text{FPT}}^{*}$ and $\mathcal{B}_{\text{QAT}}^{*}$ describe the optimal allocation of a computational budget $\mathcal{B}_{\text{train}}$. We utilize $\mathcal{B}_{\text{train}}$ to incorporate training tokens utilization ($\mathcal{D}$) into the training strategy ($\mathcal{S}$). Specifically, we evaluate various allocation ratios of $\mathcal{B}_{\text{FPT}}$ and $\mathcal{B}_{\text{QAT}}$ on MobileLLM-125M across four bit-widths ( 1.58-bit, 2-bit, 3-bit, and 4-bit). The FP models undergo a complete learning rate scheduling cycle for $\mathcal{B}_{\text{FPT}}$ tokens, followed by another cycle for QAT for $\mathcal{B}_{\text{QAT}}$ tokens. Detailed experimental settings are provided in the appendix.

Figure 2 reveals a distinct upward trend in the full-precision pre-training proportion versus accuracy curve. Notably, accuracy peaks at $\sim 90\%$ FPT allocation for almost every bit-width choice, then decline sharply when FPT exceeds 90%, likely because this leaves insufficient tokens and training capacity for QAT. This leads to our first key finding:

> **Finding-1** QAT finetuning consistently surpasses both PTQ with $\mathcal{B}_{\text{FPT}} = \mathcal{B}_{\text{train}}$ and QAT from scratch with $\mathcal{B}_{\text{QAT}} = \mathcal{B}_{\text{train}}$. Optimal performance is nearly achieved by dedicating the majority of the training budget to full precision (FP) training and approximately 10% to QAT.

## 2.2 Finetuning Characteristics

Then we investigate the impact of finetuning tokens across various bit choices, spanning 7 architectures and 5 bit levels. Results in Figure 3 offer several key insights:

1. **finetuning benefits across all bit-widths**: This observation challenges recent methodologies that trains ternary LLMs from scratch [Kaushal et al., 2024, Ma et al., 2024]. Instead, we suggest leveraging pre-trained full-precision models for initialization is a more effective approach for training quantized networks, including binary and ternary.

2. **Optimal finetuning budget and bit width**: Lower bit quantization (binary, ternary, 2-bit) requires more finetuning than higher bit quantization (3-bit, 4-bit). 3-bit and 4-bit reach near full precision accuracy after 10B tokens, while lower-bit quantization saturates around 30B tokens.

3. **QAT behavior transition between bit-widths**: Networks quantized to 3-bit/4-bit recover near full-precision accuracy after finetuning, while binary, ternary, and 2-bit saturate before achieving full accuracy. We hypothesize that QAT acts as "*compensation*" for bit-widths above 2-bit, adjusting weights within adjacent quantization levels, and as "*reconstruction*" below 2-bit, where weights adapt beyond nearby grids to form new representations. This is supported by weight change analysis in Figure 4, showing smaller adjustments in 3-bit/4-bit (10-20%) and larger shifts in lower-bit quantization ($\sim 40\%$), indicating substantial value reconstruction.

> **Finding-2** While finetuning enhances performance across all bit-widths, even binary and ternary, optimal finetuning effort inversely correlates with bit-width. For 3- and 4-bit weights, finetuning adjusts within a nearby grid to mitigate accuracy loss, and requires less finetuning tokens. In contrast, binary and ternary weights break the grid, creating new semantic representations to maintain performance, requiring longer finetuning.

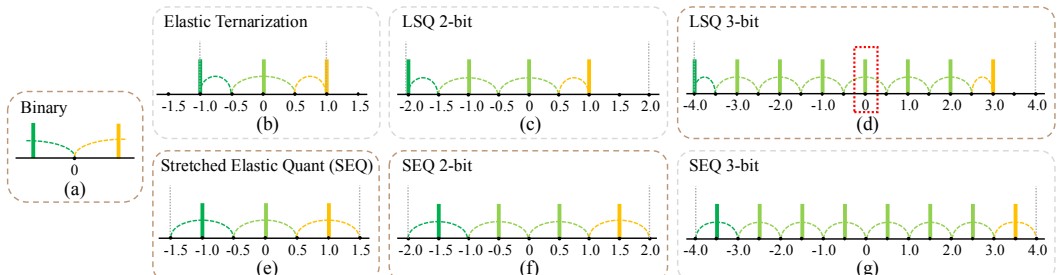

Figure 5: Impact of quantization grid choice across bit widths. Binary quantization uses a sign function; Ternary and 2-bit prefer more balanced output levels and range coverage; For 3-bit and higher, including "0" in quantization levels is more favorable.

# 3 A Hitchhiker's Guide to Quantization Method Choices

We have examined the impact of training strategy and budget allocations ($\mathcal{B}_{\text{train}}$, $\mathcal{B}_{\text{QAT}}$) on scaling laws. Building on the optimal training practices outlined in Section 2, we focus on a critical yet often overlooked factor: the choice of quantization functions ($\mathcal{F}$).

$$\mathcal{F}^* = \arg\min_{\mathcal{F}} \mathcal{L}(\mathcal{F}|\mathcal{P}, \mathcal{B}_{\text{QAT}}^*) \tag{2}$$

The significance of this choice has been largely underestimated in prior scaling law studies [Kumar et al., 2024]. Our results show that, especially at sub-4-bit quantization, the choice of function is highly sensitive and can drastically alter scaling law outcomes. An improper selection can distort performance and lead to entirely different conclusions, underscoring the need for careful design of $\mathcal{F}$.

## 3.1 Preliminary

In general, a uniform quantization function is expressed as $\mathbf{W}_{\mathbf{Q}}^i = \alpha \lfloor \frac{\mathbf{W}_{\mathbf{R}}^i - \beta}{\alpha} \rceil + \beta$, where $\mathbf{W}_{\mathbf{Q}}$ represents quantized weights, $\mathbf{W}_{\mathbf{R}}$ denotes their real-valued counterparts [Nagel et al., 2021, Krishnamoorthi, 2018]. Key design choices focus on scale $\alpha$ and bias $\beta$. For symmetric min-max quantization, $\alpha = \frac{\max(|\mathbf{W}_{\mathbf{R}}|)}{2^{N-1}-1}$ and $\beta = 0$. In asymmetric min-max quantization, $\alpha = \frac{\max(\mathbf{W}_{\mathbf{R}}) - \min(\mathbf{W}_{\mathbf{R}})}{2^N - 1}$ and $\beta = \min(\mathbf{W}_{\mathbf{R}})$. Symmetric min-max quantization is prevalent for weights $\geqslant 4$ bits, while sub-4-bit quantization requires distinct functions.

For binary quantization, assigning the sign of full-precision weights ($\mathbf{W}_{\mathbf{R}}$) to binary weights ($\mathbf{W}_{\mathbf{B}}$) is a commonly used approach [Rastegari et al., 2016, Liu et al., 2018]: $\mathbf{W}_{\mathbf{B}}^i = \alpha \cdot \text{Sign}(\mathbf{W}_{\mathbf{R}}^i)$, where $\alpha = \frac{||\mathbf{W}_{\mathbf{R}}||_{l1}}{n_{\mathbf{W}_{\mathbf{R}}}}$. In ternary quantization, ternary weights are often given by $\mathbf{W}_{\mathbf{T}}^i = \alpha \cdot \text{Sign}(\mathbf{W}_{\mathbf{R}}^i) \cdot \mathbf{1}_{|\mathbf{W}_{\mathbf{R}}^i| > \Delta}$, with $\Delta = \frac{0.7 \cdot ||\mathbf{W}_{\mathbf{R}}||_{l1}}{n_{\mathbf{W}_{\mathbf{R}}}}$ and $\alpha_{\mathbf{T}} = \frac{\sum_i \mathbf{W}_{\mathbf{R}}^i \cdot \mathbf{1}_{|\mathbf{W}_{\mathbf{R}}^i| > \Delta}}{\sum_i \mathbf{1}_{|\mathbf{W}_{\mathbf{R}}^i| > \Delta}}$ [Zhang et al., 2020, Liu et al., 2023a]. Besides binary and ternary quantization, there is less work targeting 2-bit or 3-bit integer quantization function design. Directly using min-max quantization for them will lead to performance collapse.

## 3.2 Introducing `ParetoQ`

In sub-4-bit quantization, design requirements vary significantly across bit levels. Equal attention to each bit choice is crucial for accurate, reliable comparisons.

### 3.2.1 Trade-offs

We identify two key trade-offs in low-bit quantization for LLMs: (1) Outlier precision vs. intermediate value precision and (2) Symmetry vs. inclusion of "0" at the output level.

**(1) Range clipping** Outliers challenge LLM quantization [Lin et al., 2023, Liu et al., 2024a], especially when using *min-max* ranges for weight quantization for extremely low-bit quantization. As seen in Figure 6 (b)-(e), *min-max* quantization works at 4 bits but loses accuracy at lower bit-widths. On the other hand, range clipping improves lower-bit quantization but harms 4-bit accuracy. We refer

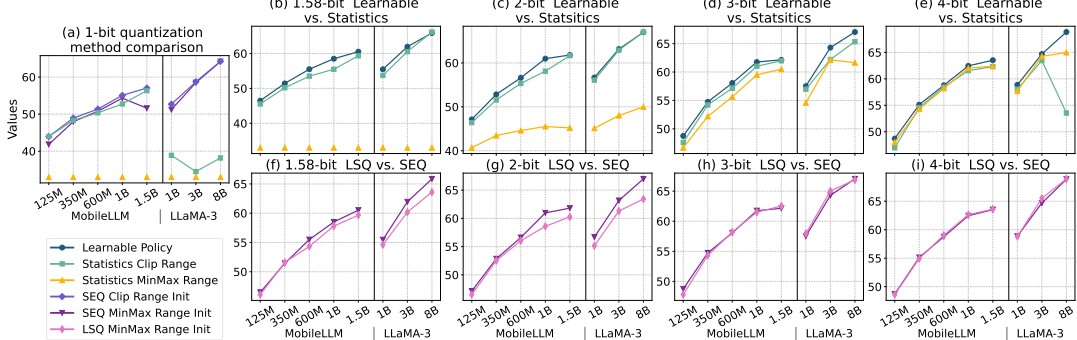

Figure 6: Comparison of quantization methods across different bit-widths. Extreme low-bit quantization is highly sensitive to quantization function selection. (b)-(e) show that the learnable policy with range clipping updated via final loss consistently outperforms stats-based methods with fixed range clipping. From (f)-(i), the SEQ works better for ternary and 2-bit quantization, while 3 and 4-bits favor LSQ.

to range-setting methods based on weight statistics as "stats-based" approaches. The effectiveness of these quantization functions varies with different bit choices.

Learnable scales, however, optimize quantization ranges as network parameters, balancing outlier suppression and precision. Solutions like LSQ [Esser et al., 2019] and its binary [Liu et al., 2022] and ternary [Liu et al., 2023a] extensions exist. While prior work favored learnable policies for activations but used statistics-based quantization for weights [Liu et al., 2023b], we find that, with appropriate gradient scaling, learnable scales yield stable, superior performance for weights. As shown in Figure 6 (b)-(e), learnable policies consistently outperform stats-based methods across all bit widths.

**(2) Quantization grids** Level symmetry in quantization grids is crucial for lower-bit quantization, yet it is rarely discussed. The "0" in quantization output levels is essential for nullifying irrelevant information, but in even-level quantization (e.g., 2-bit, 3-bit, 4-bit), including "0" results in imbalanced levels. For example, in 2-bit quantization, options like $(-2, -1, 0, 1)$ and $(-1.5, -0.5, 0.5, 1.5)$ exist. The former limits representation with only one positive level, while the latter offers a balanced distribution. Inspired by this, we propose Stretched Elastic Quant (SEQ), an amendment to LSQ for lower-bit scenarios: $\mathbf{W}_{\mathbf{Q}}^i = \alpha \left( \lfloor \text{Clip} \left( \frac{\mathbf{W}_{\mathbf{R}}^i}{\alpha}, -1, 1 \right) \times \frac{k}{2} - 0.5 \rceil + 0.5 \right) / k \times 2$. Here, $k$ denotes the number of quantization levels. Figure 5 visualizes quantization grids, showing that SEQ not only balances output quantized levels but also evenly divides the full-precision weight span to quantization levels, which turns out to be crucial for extremely low-bit quantization. Figure 6 (f)-(i) demonstrate SEQ's superiority in ternary and 2-bit quantization, while LSQ with "0" in output level slightly outperforms in 3 and 4-bit cases.

> **Finding-3** Extreme low-bit quantization is highly sensitive to quantization function selection, with no single optimal function for all bit widths. Learnable range settings outperform statistics-based methods due to their flexibility in optimizing range parameters with respect to the final loss. Ternary and 2-bit quantization favor symmetric levels and balanced range coverage in quantization grid configuration, while imbalance levels with "0" in output levels are more effective for 3 and 4-bit quantization.

### 3.2.2 Quantization Function

Based on our analysis, we integrate the optimal quantization functions identified for each bit-width into one formula, denoted as `ParetoQ`. This includes Elastic Binarization [Liu et al., 2022] for 1-bit quantization, LSQ [Esser et al., 2019] for 3 and 4-bit quantization, and the proposed SEQ for 1.58 and 2-bit quantization:

$$\mathbf{W}_{\mathbf{Q}}^i = \alpha \hat{\mathbf{W}}_{\mathbf{Q}}^i = \begin{cases} \alpha \cdot \text{Sign}(\mathbf{W}_{\mathbf{R}}^i), & \text{if } N_{bit} = 1 \\ \alpha ( \lfloor \text{Clip}(\frac{\mathbf{W}_{\mathbf{R}}^i}{\alpha}, -1, 1) \times k/2 - 0.5 \rceil + 0.5 ) / k \times 2, & \text{if } N_{bit} = 1.58, 2 \\ \alpha \lfloor \text{Clip}(\frac{\mathbf{W}_{\mathbf{R}}^i}{\alpha}, n, p) \rceil, & \text{if } N_{bit} = 3, 4 \end{cases} \quad (3)$$

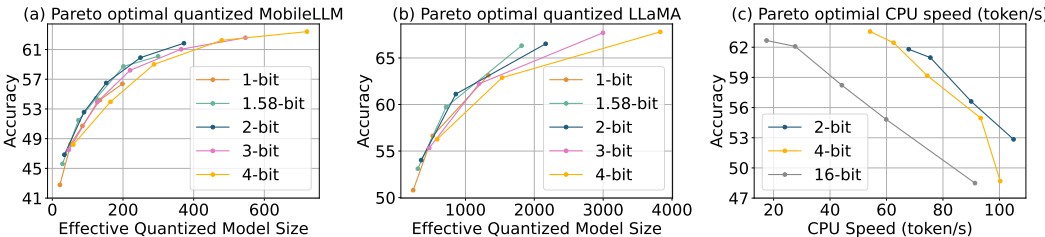

Figure 7: (a) (b) In sub-4-bit regime, 1.58-bit, 2-bit, and 3-bit quantization outperform 4-bit in terms of the accuracy-model size trade-off. (c) Under hardware constraints, 2-bit quantization demonstrates superior accuracy-speed trade-offs compared to higher-bit schemes.

Here $k$ equals 3 in the ternary case and $2^{N_{bit}}$ otherwise; $n = -2^{N_{bit}-1}$ and $p = 2^{N_{bit}-1} - 1$. In the backward pass, the gradients to the weights and scaling factor can be easily calculated using straight-through estimator:

$$\frac{\partial \mathbf{W}_{\mathbf{Q}}^i}{\partial \mathbf{W}_{\mathbf{R}}^i} \overset{STE}{\approx} \begin{cases} \mathbf{1}_{|\frac{\mathbf{W}_{\mathbf{R}}^i}{\alpha}|<1}, & \text{if } N_{bit}=1, 1.58, 2 \\ \mathbf{1}_{n<\frac{\mathbf{W}_{\mathbf{R}}^i}{\alpha}<p}, & \text{if } N_{bit}=3, 4 \end{cases} \qquad \frac{\partial \mathbf{W}_{\mathbf{Q}}^i}{\alpha} \overset{STE}{\approx} \begin{cases} \text{Sign}(\mathbf{W}_{\mathbf{R}}^i), & \text{if } N_{bit}=1 \\ \hat{\mathbf{W}}_R^i - \frac{\mathbf{W}_{\mathbf{R}}^i}{\alpha} \cdot \mathbf{1}_{|\frac{\mathbf{W}_{\mathbf{R}}^i}{\alpha}|<1}, & \text{if } N_{bit}=1.58, 2 \\ \hat{\mathbf{W}}_R^i - \frac{\mathbf{W}_{\mathbf{R}}^i}{\alpha} \cdot \mathbf{1}_{n<\frac{\mathbf{W}_{\mathbf{R}}^i}{\alpha}<p}, & \text{if } N_{bit}=3, 4 \end{cases}$$

$$(4) \qquad\qquad\qquad\qquad\qquad\qquad\qquad (5)$$

For the initialization of $\alpha$, we use $\alpha = \frac{||\mathbf{W}_{\mathbf{R}}||_{l1}}{n_{\mathbf{W}_{\mathbf{R}}}}$ for the binary case, since the scaling factor has the closed-form solution to minimizing quantization error: $\mathcal{E} = ||\alpha \hat{\mathbf{W}}_{\mathbf{Q}} - \mathbf{W}_{\mathbf{R}}||_{l2}$. For the other cases, we simply initialize $\alpha$ as the maximum absolute value of the weights. For ternary and 2-bit quantization, $\alpha = \max(|\mathbf{W}_{\mathbf{R}}|)$, associated with SEQ quantizer, and for 3-bit and 4-bit cases, $\alpha = \frac{\max(|\mathbf{W}_{\mathbf{R}}|)}{p}$, associated with LSQ quantizer.

With `ParetoQ`, we present a robust comparison framework across five bit-widths (1-bit, 1.58-bit, 2-bit, 3-bit, 4-bit), each achieving state-of-the-art accuracy. This facilitates direct, apple-to-apple comparisons to identify the most effective bit-width selection.

## 4   Pareto-Optimality of Extremely Low-Bit LLM

To ensure a consistent apples-to-apples performance comparison across different bit-width configurations, we first determined the optimal training setup ($\mathcal{B}_{train}^*$) in Section 2 and the quantization function ($\mathcal{F}^*$) for each bit in Section 3. Using this unified framework for all bit widths, we examine the trade-off between model size and quantization bit: $\mathcal{L}(\mathcal{P}, \mathcal{N} | \mathcal{F}^*, \mathcal{B}_{train}^*)$.

### 4.1   Accuracy-compression Trade-off

In on-device deployment scenarios, such as wearables and portables, storage constraints often limit the capacity of large language models (LLMs). To optimize performance within these constraints, quantization is essential. A common dilemma is whether to train a larger model and quantize it to a lower bit-width or to train a smaller model and quantize it to a higher bit-width.

4-bit quantization-aware training (QAT) achieves near-lossless compression in many scenarios, making it widely adopted. However, the landscape below 4-bit remains unclear, with limited comparative analysis. Previous claims about ternary models matching 16-bit performance [Ma et al., 2024] were based on lower FP16 baselines than current standards. Spectra's comparisons between ternary QAT and 4-bit PTQ fall short of a fair evaluation due to inconsistencies in the training schemes used [Kaushal et al., 2024].

With `ParetoQ`, we are able to improve the analysis. Figure 7 (a) demonstrates that sub-4-bit quantization, including binary, ternary, 2-bit, and 3-bit, often surpasses 4-bit. Notably, 2-bit and ternary models reside on the Pareto frontier. For instance, a 2-bit MobileLLM-1B model achieves 1.8 points higher accuracy than a 4-bit MobileLLM-600M model, with even smaller model sizes. This trend persists across larger LLaMA models, as shown in Figure 7 (b), demonstrating the potential of lower-bit quantization for achieving both higher accuracy and compression. We calculate the effective

quantized model size as $(\#\text{weights} \times \text{weight-bits} + \#\text{embedding-weights} \times \text{embedding-bits})/8$. More comprehensive analysis is provided in the Appendix.

## 4.2 Hardware Implementation Constraints

In practical deployment, both memory limitations and hardware constraints must be considered. While 2-bit and ternary quantization sit on the accuracy-size Pareto frontier, 2-bit quantization is generally more feasible due to practical challenges. Ternary quantization, using a 1.58-bit format with values $\{-1, 0, 1\}$, appears more storage-efficient but is inefficient in implementation. Storing ternary values with sparsity exploitation is effective only when sparsity exceeds 90%, due to high indexing costs. Packing ternary values into an Int32 offers limited compression but complicates GEMM. Some approaches [Yang et al., 2024] even store ternary values as 2-bit signed integers, negating the expected storage benefits. In contrast, 2-bit quantization directly maps bit pairs to values, reducing unpacking and conversion overhead, which can be more efficient for custom GEMM kernels. As a result, 2-bit quantization is often a more practical choice for deployment.

## 4.3 Accuracy-speed Trade-off

To evaluate potential speedup benefits beyond memory reduction, we implemented 2-bit quantization kernels on the CPU and compared them with 4-bit quantization. The curves in Figure 7 (c) demonstrate that, within our experimental range, 2-bit quantized models consistently outperform 4-bit models in terms of accuracy-speed performance, positioning 2-bit quantization as a superior choice for on-device applications where both latency and storage are critical. See appendix for detailed settings.

# 5 Experiments

In this section, we compare each point on our Pareto chart with prior methods in the literature. As the first approach to unify training and quantization schemes in the sub-4-bit regime, we evaluate our method against specialized techniques for each bit setting. This includes binary quantization methods: BiLLM [Huang et al., 2024], ARB-LLM [Li et al., 2024], PB-LLM [Shang et al., 2023], and DB-LLM [Chen et al., 2024a]; ternary quantization methods: TernaryLLM [Chen et al., 2024c], 1-bit Era [Ma et al., 2024]; and lower-bit QAT methods: LLM-QAT [Liu et al., 2023c] and EfficientQAT [Chen et al., 2024b] as well as PTQ methods like GPTQ [Frantar et al., 2022], OmniQ [Shao et al., 2023], SpinQuant [Liu et al., 2024a], QuIP [Chee et al., 2024] and AWQ [Lin et al., 2023]. We also compare with a post-training vector quantization method AQLM [Egiazarian et al., 2024].

We demonstrate that `ParetoQ`, with a unified scheme spanning five distinct bit settings (1, 1.58, 2, 3, and 4 bits), consistently outperforms previous methods specialized for each bit level, including both PTQ and QAT approaches. The performance gains are particularly pronounced in the 1, 1.58, and 2-bit settings, underscoring the robustness and reliability of our conclusions regarding scaling laws.

## 5.1 Experimental Settings

We conduct experiments on eight models including MobileLLM [Liu et al., 2024b] 125M/350M/600M/1B/1.5B and LLaMA-3 [AI@Meta, 2024] 1B/3B/8B. Our evaluation was carried out on eight zero-shot commonsense reasoning tasks and Wiki2 [Merity et al., 2016] test set.

During the quantized network training process, we initialized the models with pre-trained weights. Following the common practice [Frantar et al., 2022, Liu et al., 2023c], all weights except for the embedding and output layers are quantized. We employed the AdamW [Loshchilov and Hutter, 2017] optimizer with zero weight decay for optimization. The training was distributed across 16 GPUs, with each GPU handling a batch size of 8. For binary, ternary, and 2-bit quantization settings, the optimization process spanned 120,000 iterations with initial learning rate of $2 \times 10^{-5}$. For 3-bit and 4-bit settings, the process involved 40,000 iterations with initial learning rate of $1 \times 10^{-5}$. The learning rate decayed to zero following cosine learning rate decay.

## 5.2 Main Results

**1 / 1.58 / 2-bit Comparison on 8B Model**  Let's first examine the comparison on 8B parameter models. As depicted in Table 1, in the 2-bit quantization setting, previous methods, including both PTQ and QAT, experience a significant drop in accuracy. Among PTQ methods, the vector quantization method

Table 1: Comparison of 1-bit, 1.58-bit and 2-bit quantization methods on LLaMA-3 8B model. Results for LLM-QAT, EfficientQAT, GPTQ, AWQ, SpinQuant, OmniQ, AQLM, 1-bit era and BiLLM were obtained using their publicly released models and codebase. The results of DB-LLM, PB-LLM, QuIP and TernaryLLM are quoted from the TernaryLLM paper. The results of ARB-LLM is sourced from their paper. All methods employ integer quantization, except AQLM, which uses vector quantization with a vector dimension of 16.

| Method | #Bits | Group Size | ARC-e (↑) | ARC-c (↑) | PIQA (↑) | HellaS (↑) | WinoG (↑) | Avg. (↑) | Wiki2 (↓) |
|---|---|---|---|---|---|---|---|---|---|
| FP | 16 | – | 81.0 | 57.7 | 81.0 | 79.5 | 73.9 | 74.6 | 6.15 |
| RTN | 2 | channel | 27.2 | 25.1 | 49.7 | 26.1 | 50.5 | 35.7 | 1.2e6 |
| GPTQ | 2 | channel | 27.4 | 24.6 | 51.0 | 25.9 | 50.6 | 35.9 | 1.6e2 |
| OmniQ | 2 | channel | 27.3 | 22.8 | 49.5 | 25.3 | 49.4 | 34.8 | – |
| SpinQuant | 2 | channel | 32.4 | 21.8 | 53.4 | 31.9 | 50.9 | 38.1 | 31.2 |
| AWQ | 2 | channel | 26.0 | 27.1 | 51.4 | 26.1 | 49.8 | 36.1 | – |
| QuIP | 2 | channel | – | – | – | – | – | – | 85.1 |
| AQLM | 2.27 | 1x16 | 75.5 | 51.8 | 78.8 | 75.3 | 69.8 | 70.2 | – |
| DB-LLM | 2.12 | 128 | – | – | – | – | – | – | 13.6 |
| PB-LLM | 2.12 | 128 | – | – | – | – | – | – | 24.7 |
| LLM-QAT | 2 | channel | 54.8 | 35.9 | 68.0 | 58.0 | 54.7 | 54.3 | 29.5 |
| EfficientQAT | 2.12 | 128 | 69.3 | 46.8 | 76.4 | 69.0 | 66.3 | 65.5 | 9.6 |
| ParetoQ | 2 | channel | **78.5** | **54.5** | **79.2** | **73.8** | 70.0 | **71.2** | **8.0** |
| PB-LLM | 1.7 | 128 | – | – | – | – | – | – | 41.8 |
| 1-bit era | 1.58 | channel | 72.8 | 45.4 | **81.0** | 70.6 | 58 | 65.6 | 11.7 |
| TernaryLLM | 1.58 | channel | – | – | – | – | – | – | 11.2 |
| ParetoQ | 1.58 | channel | **76.3** | **51.4** | 77.7 | **71.9** | **67.7** | **69.0** | **8.6** |
| BiLLM | 1.06 | 128 | 33.2 | 25.6 | 54.6 | 32.7 | 50.5 | 39.3 | 38.5 |
| ARB-LLM | 1.06 | channel | – | – | – | – | – | – | 27.4 |
| ParetoQ | 1 | channel | **75.5** | **51.9** | **76.6** | **69.4** | **65.6** | **67.8** | **9.5** |

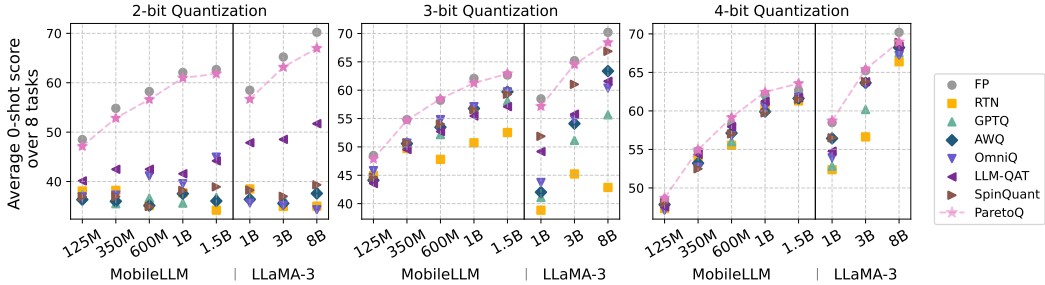

Figure 8: Accuracy comparison on eight models. ParetoQ outperforms all state-of-the-art PTQ and QAT methods.

AQLM [Egiazarian et al., 2024] effectively mitigates some of the quantization loss, achieving 64.1 points, it falls 10.5 points short of full precision. The best quantization-aware training method, EfficientQAT [Chen et al., 2024b], still suffers a 9.1-point decline in average accuracy. ParetoQ dramatically narrows the 2-bit quantization gap to full precision to just 3.4 points, outperforming the best QAT method by 5.7 points and the vector quantization method by 7.1 points.

In ternary cases, the accuracy drop is more pronounced, highlighting the effectiveness of different quantization methods. A follow-up work of the 1-bit Era [Mekkouri et al., 2024], which trains 1-bit LLaMA-3 8B models using 100B tokens and complex techniques like binary relax with sigmoid schedulers, still experiences a 9.0-point accuracy drop. In contrast, ParetoQ requiring only 30B tokens and utilizing standard AdamW optimization with cosine learning rate decay, narrows the gap to just 5.6 points. This underscores the robustness of our quantization function design.

Furthermore, ParetoQ significantly outperforms previous binary quantization techniques, such as BiLLM and ARB-LLM, reducing WikiText perplexity from 27.4 to 9.5.

**1.58-bit Comparison on Sub-8B Models** Figure 9 illustrates that ParetoQ also excels in sub-8B models, consistently outperforming previous methods targeting at ternary quantization aware training such as 1-bit Era [Ma et al., 2024]. Given that a full-precision LLaMA-3 3B model achieves 65.2 accuracy, it's remarkable that ParetoQ ternary 3B-parameter model narrows the gap to just 3.3 points, while previous methods experience drops exceeding 7.4 points. Additionally, our 600M-parameter ternary model achieves 58.7 accuracy, even surpassing 1-bit era ternary 3B model with only 1/5 parameters.

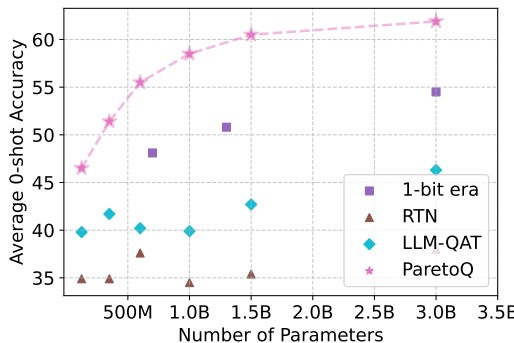

Figure 9: Ternary quantization accuracy averaged across eight zero-shot commonsense reasoning tasks. `ParetoQ` consistently outperforms all prior methods in ternary quantization-aware training.

**2-bit / 3-bit / 4-bit Comparisons**  As evidenced by Figure 8, compared to previous state-of-the-art PTQ and QAT methods on 2, 3 or 4-bit quantization settings, our approach consistently resides on the Pareto front, with a particularly pronounced advantage in lower-bit quantization settings. These results confirm that our bit-accuracy trade-off conclusions are benchmarked against SoTA results across all bit settings, ensuring its reliability.

## 6   Related Work

The quantization of Large Language Models (LLMs) has emerged as a pivotal research area, driven by the imperative to reduce computational and memory demands while preserving model performance [Liu et al., 2023c, Dettmers et al., 2022, Xiao et al., 2022]. A notable trend is the quantization of LLMs to lower bit-widths [Ma et al., 2024, Kaushal et al., 2024].

Initial efforts, such as LLM.int8() [Dettmers et al., 2022] and SmoothQuant [Xiao et al., 2022], concentrated on quantizing LLMs to 8-bit weights and 8-bit activations. Subsequently, numerous studies have demonstrated the feasibility of quantizing LLMs to 4-bit with minimal accuracy degradation, employing both post-training quantization (PTQ) methods [Kim et al., 2023, Frantar et al., 2022, Liu et al., 2024a, 2023b] and quantization-aware training (QAT) [Liu et al., 2023c, Chen et al., 2024b, Bondarenko et al., 2021].

Recently, research has shifted towards sub-4-bit quantization. Some PTQ methods target 3-bit or 2-bit integer quantization [Shao et al., 2023, Zhao et al., 2023, Chee et al., 2024, Ashkboos et al., 2023, Lin et al., 2023, Frantar et al., 2022], or employ vector quantization [Egiazarian et al., 2024, Tseng et al., 2024, van Baalen et al., 2023]. Other PTQ approaches even achieve binary weight quantization [Huang et al., 2024, Shang et al., 2023, Chen et al., 2024a, Li et al., 2024]. Most recently, two QAT studies have claimed that ternary quantized models, trained from scratch, can match the accuracy of full-precision models with equivalent training [Ma et al., 2024, Kaushal et al., 2024]. It generated heated debate within the field, with many practitioners expressing reservations about this conclusion. To our knowledge, no existing work unifies sub-4-bit quantization schemes to derive a solid conclusion on which bit-width achieves the Pareto optimal in the efficiency-accuracy trade-off. This work presents `ParetoQ` to fill that gap.

## 7   Conclusions

In this study, we have performed an in-depth analysis of the intricate relationships among model parameters ($N$), training data volume ($D$), quantization training schemes ($\mathcal{B}_{\text{train}}$), quantization precision ($P$), and the selection of quantization functions ($\mathcal{F}$) in relation to the model's final loss, expressed as $\mathcal{L} = f(N, D, P, \mathcal{B}_{\text{train}}, \mathcal{F})$. To address these multifaceted challenges, we propose `ParetoQ`, an advanced quantization framework that achieves state-of-the-art performance across all bit-width levels. This framework uniquely enables a direct, consistent comparison across different bit-widths, ensuring an equitable evaluation of performance metrics. Our empirical analysis indicates that quantization at 1.58-bit, 2-bit, and 3-bit offers a superior trade-off between accuracy and effective quantized model size compared to 4-bit, highlighting their potential for optimized model deployment.

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

# A  Appendix / supplemental material

## A.1   Complete Results of Figure 9

Table 2 presents the numerical results of Figure 9. We evaluate accuracy across eight zero-shot commonsense reasoning tasks: ARC-easy, ARC-challenge [Clark et al., 2018], BoolQ [Clark et al., 2019], PIQA [Bisk et al., 2020], SIQA [Sap et al., 2019], HellaSwag [Zellers et al., 2019], OBQA [Mihaylov et al., 2018], and WinoGrande [Sakaguchi et al., 2021], along with perplexity on the WikiText2 test set [Merity et al., 2016]. Our results are compared against prior state-of-the-art ternary quantization-aware training works, such as 1-bit era [Ma et al., 2024]. We also include the comparison to LLM-QAT [Liu et al., 2023c]. Consistent with previous methodologies [Ma et al., 2024], we quantize all weights to low-bit, excluding the embedding and output layers. The `ParetoQ` 3B ternary model is quantized from LLaMA3 [AI@Meta, 2024] 3B model, while other models are quantized from MobileLLM [Liu et al., 2024b].

Table 2: Comparison of `ParetoQ` ternary quantization with QAT methods, including general LLM-QAT [Liu et al., 2023c] and ternary-specific QAT methods such as 1-bit Era [Ma et al., 2024].

| Method | # Params | ARC-e (↑) | ARC-c (↑) | BoolQ (↑) | PIQA (↑) | SIQA (↑) | HellaSwag (↑) | OBQA (↑) | WinoGrande (↑) | Avg. (↑) | Wiki2 (↓) |
|---|---|---|---|---|---|---|---|---|---|---|---|
| RTN | 125M | 25.5 | 26.5 | 37.8 | 49.6 | 36.3 | 26.3 | 27.7 | 49.3 | 34.9 | 4.0e5 |
| LLM-QAT | 125M | 34.9 | 20.4 | 59.0 | 54.6 | 39.0 | 29.1 | 30.2 | 50.9 | 39.8 | 87.3 |
| ParetoQ | 125M | **49.3** | **30.9** | **61.0** | **62.1** | **41.0** | **34.3** | **40.4** | **52.9** | **46.5** | **19.8** |
| RTN | 350M | 26.6 | 25.1 | 37.8 | 48.7 | 36.7 | 26.5 | 27.5 | 50.2 | 34.9 | 3.7e5 |
| LLM-QAT | 350M | 39.1 | 24.1 | 61.6 | 55.5 | 39.9 | 30.4 | 32.1 | 50.6 | 41.7 | 68.6 |
| ParetoQ | 350M | **56.8** | **36.3** | **62.2** | **67.1** | **43.5** | **44.0** | **46.3** | **55.2** | **51.4** | **14.4** |
| RTN | 600M | 26.2 | 24.6 | 62.2 | 49.5 | 36.3 | 26.1 | 27.1 | 48.8 | 37.6 | 6.6e5 |
| LLM-QAT | 600M | 34.0 | 23.0 | 59.4 | 53.6 | 38.9 | 28.7 | 32.3 | 51.4 | 40.2 | 71.7 |
| 1-bit era | 700M | 49.5 | 29.0 | 59.2 | 67.5 | 43.6 | 43.2 | 38.9 | 53.5 | 48.0 | 17.3 |
| ParetoQ | 600M | **65.5** | **43.8** | **62.3** | **70.6** | **44.7** | **51.3** | **47.1** | **58.8** | **55.5** | **11.4** |
| RTN | 1B | 25.7 | 24.8 | 37.8 | 49.3 | 37.1 | 26.2 | 25.2 | 50.2 | 34.5 | 1.4e5 |
| LLM-QAT | 1B | 36.0 | 26.2 | 47.7 | 55.1 | 39.7 | 31.3 | 33.5 | 49.6 | 39.9 | 56.9 |
| 1-bit era | 1.3B | 52.4 | 34.1 | 61.9 | 69.1 | 44.7 | 47.4 | 41.1 | 55.3 | 50.8 | 23.6 |
| ParetoQ | 1B | **68.5** | **47.6** | **62.8** | **72.1** | **45.3** | **57.4** | **52.9** | **61.3** | **58.5** | **10.0** |
| RTN | 1.5B | 25.5 | 26.8 | 37.8 | 49.0 | 37.6 | 26.0 | 30.5 | 50.2 | 35.4 | 9.7e4 |
| LLM-QAT | 1.5B | 41.1 | 26.1 | 60.5 | 57.6 | 39.5 | 35.0 | 31.9 | 49.8 | 42.7 | 39.7 |
| ParetoQ | 1.5B | **70.2** | **48.0** | **65.8** | **73.4** | **47.3** | **61.8** | **55.3** | **62.4** | **60.5** | **9.0** |
| RTN | 3B | 26.9 | 23.6 | 62.2 | 51.3 | 37.6 | 26.4 | 27.0 | 49.3 | 38.0 | 4.4e5 |
| LLM-QAT | 3B | 44.5 | 30.7 | 62.1 | 62.7 | 41.0 | 43.4 | 35.0 | 50.6 | 46.3 | 6.5e2 |
| 1-bit era | 3B | 58.7 | 37.2 | 61.3 | 71.3 | 45.2 | 56.0 | 45.8 | 60.3 | 54.5 | 265.6 |
| ParetoQ | 3B | **71.5** | **48.6** | **68.2** | **75.5** | **46.4** | **67.9** | **54.3** | **63.1** | **61.9** | **9.9** |

## A.2   Complete Results of Figure 8

In Tables 3, 4, and 5, we provide detailed results corresponding to Figure 8. We compare `ParetoQ` against LLM-QAT [Liu et al., 2023c], GPTQ [Frantar et al., 2022], AWQ [Lin et al., 2023], OmniQuant [Shao et al., 2023], and SpinQuant [Liu et al., 2024a]. Following the common practice [Frantar et al., 2022, Liu et al., 2023c], we apply low-bit quantization to all weights, except for the embedding and output layers.

## A.3   CPU Latency Experimental Setup

We measure the CPU latency of five MobileLLM models on an Apple M1 MacBook Pro (32GB RAM) using 6 threads. Each evaluation uses 5 prompt tokens and generates 122 tokens. For the quantized models, embedding and output layers are quantized to 8-bit precision using channelwise quantization, while weights in fully connected layers are quantized to 2-bit or 4-bit precision. Accuracy and decoding speed (in tokens/s) were measured under identical settings.

## A.4   GPU Latency Experimental Setup and Results

We measured the latency of LLaMA 3.2 models (1B, 3B, 8B) on an H100 NVL GPU (94GB memory). The W4A16 kernel used the Machete kernel from vLLM [Kwon et al., 2023, Wilkinson, 2024], while the W2A16 kernel was implemented based on the CUTLASS mixed precision backbone kernel. All tests were performed on a single GPU with a context length of 2048 tokens. For kernel-level latency, we compared the 2-bit kernel to the 4-bit Machete kernel across three weight shapes: (4096 × 4096), (8192 × 8192), and (16384 × 16384).

Table 3: Complete results of **2-bit quantization** on WikiText2 and Zero-shot Common Sense Reasoning tasks.

| Model Name | Method | ARC-e (↑) | ARC-c (↑) | BoolQ (↑) | PIQA (↑) | SIQA (↑) | HellaSwag (↑) | OBQA (↑) | WinoGrande (↑) | Avg. (↑) | Wiki2 (↓) |
|---|---|---|---|---|---|---|---|---|---|---|---|
| MobileLLM-125M | FP | 56.0 | 34.5 | 56.3 | 65.5 | 42.0 | 40.1 | 42.2 | 51.3 | 48.5 | 14.9 |
| | RTN | 26.1 | 24.1 | 62.2 | 50.3 | 37.1 | 26.6 | 28.9 | 49.4 | 38.1 | 6.4e5 |
| | GPTQ | 28.9 | 26.2 | 44.2 | 51.1 | 39.1 | 28.1 | 33.2 | 48.0 | 37.3 | 2.4e2 |
| | AWQ | 25.8 | 24.2 | 44.2 | 50.7 | 38.8 | 26.2 | 29.2 | 51.6 | 36.3 | 6.5e3 |
| | OmniQ | 32.4 | 22.7 | 38.1 | 53.4 | 39.4 | 28.2 | 30.9 | 49.9 | 36.9 | 1.2e2 |
| | LLM-QAT | 34.9 | 23.3 | 61.8 | 53.8 | 39.3 | 29.1 | 27.4 | 51.3 | 40.1 | 66.8 |
| | SpinQuant | 31.6 | 23.3 | 40.3 | 52.2 | 40.5 | 28.6 | 28.9 | 50.1 | 36.9 | 68.7 |
| | ParetoQ | 50.7 | 32.7 | 59.8 | 63.3 | 41.0 | 36.3 | 40.6 | 52.7 | 47.1 | 25.1 |
| MobileLLM-350M | FP | 65.5 | 42.3 | 57.4 | 71.0 | 43.5 | 53.3 | 47.3 | 58.3 | 54.8 | 10.4 |
| | RTN | 25.9 | 26.5 | 62.2 | 49.8 | 37.7 | 26.3 | 26.0 | 51.2 | 38.2 | 60.3 |
| | GPTQ | 28.6 | 21.5 | 40.5 | 50.4 | 38.8 | 26.6 | 27.3 | 50.4 | 35.5 | 1.6e2 |
| | AWQ | 27.0 | 23.5 | 47.6 | 49.4 | 38.2 | 26.4 | 26.2 | 49.5 | 36.0 | 7.2e4 |
| | OmniQ | 33.9 | 23.4 | 39.6 | 54.9 | 38.4 | 28.6 | 29.4 | 49.7 | 37.2 | 80.8 |
| | LLM-QAT | 40.6 | 25.9 | 62.0 | 55.6 | 40.0 | 31.8 | 31.1 | 52.6 | 42.5 | 8.2e4 |
| | SpinQuant | 32.4 | 25.0 | 37.8 | 54.6 | 40.1 | 29.2 | 27.5 | 48.9 | 36.9 | 67.5 |
| | ParetoQ | 59.0 | 39.4 | 63.5 | 68.8 | 43.1 | 47.3 | 44.1 | 57.5 | 52.8 | 17.7 |
| MobileLLM-600M | FP | 68.5 | 47.6 | 60.5 | 72.5 | 44.4 | 59.5 | 51.4 | 61.4 | 58.2 | 9.0 |
| | RTN | 25.8 | 26.2 | 37.8 | 49.8 | 37.6 | 25.9 | 26.8 | 50.9 | 35.1 | 2.7e2 |
| | GPTQ | 27.9 | 26.6 | 48.2 | 49.5 | 39.0 | 25.9 | 26.8 | 49.4 | 36.6 | 3.4e2 |
| | AWQ | 26.4 | 25.2 | 40.6 | 50.7 | 38.7 | 26.5 | 23.6 | 49.3 | 35.1 | 8.9e3 |
| | OmniQ | 39.0 | 24.5 | 55.8 | 55.9 | 40.2 | 30.1 | 32.1 | 51.3 | 41.1 | 68.3 |
| | LLM-QAT | 42.7 | 25.6 | 62.1 | 56.0 | 38.8 | 33.7 | 29.6 | 51.5 | 42.5 | 4.7e2 |
| | SpinQuant | 28.2 | 22.4 | 39.8 | 52.0 | 38.0 | 27.9 | 22.1 | 49.1 | 34.9 | 2.7e2 |
| | ParetoQ | 67.7 | 43.3 | 63.0 | 72.1 | 44.8 | 53.9 | 49.8 | 58.4 | 56.6 | 15.4 |
| MobileLLM-1B | FP | 73.4 | 50.8 | 67.6 | 74.1 | 46.7 | 64.7 | 56.6 | 62.7 | 62.1 | 8.0 |
| | RTN | 26.3 | 26.5 | 62.2 | 49.2 | 36.9 | 26.0 | 25.8 | 48.8 | 37.7 | 6.0e4 |
| | GPTQ | 29.7 | 25.4 | 38.7 | 50.3 | 38.9 | 26.1 | 26.4 | 49.6 | 35.6 | 4.7e2 |
| | AWQ | 26.6 | 26.8 | 59.1 | 50.2 | 37.1 | 26.0 | 24.0 | 50.4 | 37.5 | 1.5e5 |
| | OmniQ | 38.0 | 26.1 | 41.7 | 54.6 | 40.1 | 31.1 | 33.3 | 51.4 | 39.5 | 46.3 |
| | LLM-QAT | 42.6 | 26.7 | 49.7 | 57.7 | 40.4 | 34.9 | 31.4 | 49.2 | 41.6 | 1.9e5 |
| | SpinQuant | 35.3 | 23.9 | 42.8 | 53.3 | 40.5 | 30.3 | 29.7 | 49.8 | 38.2 | 35.7 |
| | ParetoQ | 73.3 | 49.3 | 65.7 | 74.2 | 45.9 | 60.3 | 57.4 | 61.6 | 61.0 | 13.4 |
| MobileLLM-1.5B | FP | 73.9 | 51.4 | 70.0 | 74.8 | 46.6 | 66.4 | 55.1 | 63.2 | 62.7 | 7.8 |
| | RTN | 25.2 | 25.3 | 37.8 | 49.3 | 36.0 | 26.4 | 25.0 | 48.5 | 34.2 | 1.7e2 |
| | GPTQ | 29.8 | 22.3 | 45.3 | 53.4 | 39.3 | 27.0 | 25.8 | 51.4 | 36.8 | 1.7e2 |
| | AWQ | 28.9 | 26.1 | 43.7 | 51.1 | 37.7 | 26.4 | 24.4 | 49.8 | 36.0 | 7.1e3 |
| | OmniQ | 50.6 | 30.6 | 54.6 | 59.7 | 40.6 | 38.9 | 32.1 | 52.2 | 44.9 | 31.3 |
| | LLM-QAT | 45.3 | 26.5 | 61.6 | 58.6 | 40.1 | 37.5 | 33.1 | 50.6 | 44.2 | 33.9 |
| | SpinQuant | 34.0 | 21.6 | 52.3 | 54.1 | 39.4 | 29.5 | 29.9 | 50.5 | 38.9 | 37.4 |
| | ParetoQ | 73.3 | 47.5 | 70.1 | 74.1 | 46.8 | 64.6 | 55.5 | 62.5 | 61.8 | 11.7 |
| LLaMA-1B | FP | 64.8 | 42.5 | 64.8 | 74.8 | 44.8 | 64.4 | 50.2 | 61.5 | 58.5 | 9.6 |
| | RTN | 26.5 | 26.8 | 62.2 | 51.0 | 36.8 | 25.9 | 28.5 | 50.2 | 38.5 | 1.5e6 |
| | GPTQ | 29.3 | 27.6 | 37.8 | 51.5 | 38.6 | 26.5 | 32.0 | 50.8 | 36.8 | 3.3e2 |
| | AWQ | 27.4 | 26.0 | 48.9 | 50.2 | 37.0 | 25.7 | 24.4 | 51.5 | 36.4 | 2.0e5 |
| | OmniQ | 27.9 | 24.7 | 39.0 | 51.1 | 40.4 | 26.0 | 26.2 | 50.0 | 35.6 | 3.3e3 |
| | LLM-QAT | 49.2 | 33.3 | 62.0 | 63.9 | 41.1 | 41.5 | 37.5 | 54.4 | 47.9 | 1.3e5 |
| | SpinQuant | 25.6 | 24.6 | 62.4 | 51.6 | 36.1 | 25.8 | 29.1 | 50.8 | 38.3 | 46.7 |
| | ParetoQ | 64.8 | 41.7 | 62.8 | 73.1 | 44.0 | 56.6 | 52.0 | 58.5 | 56.7 | 12.5 |
| LLaMA-3B | FP | 72.6 | 50.7 | 74.6 | 78.2 | 48.5 | 74.3 | 53.7 | 69.2 | 65.2 | 7.7 |
| | RTN | 26.9 | 25.1 | 37.8 | 50.1 | 37.9 | 25.7 | 26.6 | 49.6 | 35.0 | 7.8e5 |
| | GPTQ | 28.6 | 22.9 | 46.4 | 50.0 | 38.4 | 27.1 | 30.1 | 50.1 | 36.7 | 2.7e2 |
| | AWQ | 27.3 | 27.5 | 38.2 | 51.1 | 38.3 | 26.1 | 25.4 | 50.7 | 35.6 | 6.2e5 |
| | OmniQ | 28.3 | 24.6 | 37.8 | 50.5 | 38.0 | 25.3 | 26.6 | 50.2 | 35.2 | 6.5e3 |
| | LLM-QAT | 49.3 | 33.3 | 63.5 | 65.2 | 41.7 | 48.9 | 34.2 | 52.2 | 48.5 | 2.9e5 |
| | SpinQuant | 28.3 | 23.7 | 53.2 | 51.1 | 38.8 | 26.1 | 25.8 | 49.0 | 37.0 | 57.4 |
| | ParetoQ | 73.9 | 49.0 | 68.8 | 76.4 | 47.0 | 69.2 | 56.6 | 64.4 | 63.2 | 9.1 |
| LLaMA-8B | FP | 81.0 | 57.7 | 83.6 | 81.0 | 49.3 | 79.5 | 55.7 | 73.9 | 70.2 | 6.2 |
| | RTN | 27.2 | 25.1 | 37.8 | 49.7 | 37.4 | 26.1 | 26.2 | 50.5 | 35.0 | 1.2e6 |
| | GPTQ | 27.0 | 26.1 | 61.6 | 50.5 | 37.4 | 26.0 | 27.5 | 49.7 | 38.2 | 1.6e2 |
| | AWQ | 26.0 | 27.1 | 58.3 | 51.4 | 38.0 | 26.1 | 23.8 | 49.8 | 37.6 | 1.1e6 |
| | OmniQ | 27.3 | 22.8 | 37.9 | 49.5 | 38.7 | 25.3 | 23.4 | 49.4 | 34.3 | 7.6e4 |
| | LLM-QAT | 54.8 | 35.9 | 64.8 | 68.0 | 41.8 | 58.0 | 35.7 | 54.7 | 51.7 | 29.5 |
| | SpinQuant | 32.4 | 22.0 | 59.0 | 53.2 | 38.4 | 31.9 | 28.0 | 49.9 | 39.3 | 31.2 |
| | ParetoQ | 78.5 | 54.5 | 76.4 | 79.2 | 48.9 | 73.8 | 54.5 | 70.0 | 67.0 | 8.0 |

For smaller models (1B, 3B, 8B), the performance speed-up from reducing weight precision from 4-bit to 2-bit is minimal. This is due to the impact of conversion overhead, which becomes more pronounced when the weight size is small. Since the in-kernel conversion latency ratio is higher for smaller models, the benefits of 2-bit quantization are outweighed by the overhead. Consequently, 4-bit quantization achieves a more favorable speed-accuracy trade-off in these settings, offering better overall performance. In comparison, for larger weight shapes ($16384 \times 16384$), the 2-bit kernel provides a substantial speedup, achieving $4.14\times$ faster performance than FP16 and $1.24\times$ faster than the Machete 4-bit kernel.

## A.5 QAT Scheduling Experimental Setup

The total training budget ($\mathcal{B}_{\text{train}}$) is set to 100B tokens. We vary the proportion of tokens allocated for full-precision training versus quantization-aware training (QAT) finetuning, sweeping the ratio across

Table 4: Complete results of **3-bit quantization** on WikiText2 and Zero-shot Common Sense Reasoning tasks.

| Model Name | Method | ARC-e (↑) | ARC-c (↑) | BoolQ (↑) | PIQA (↑) | SIQA (↑) | HellaSwag (↑) | OBQA (↑) | WinoGrande (↑) | Avg. (↑) | Wiki2 (↓) |
|---|---|---|---|---|---|---|---|---|---|---|---|
| MobileLLM-125M | FP | 56.0 | 34.5 | 56.3 | 65.5 | 42.0 | 40.1 | 42.2 | 51.3 | 48.5 | 14.9 |
| | RTN | 45.7 | 30.0 | 59.0 | 60.5 | 40.4 | 34.9 | 38.3 | 50.5 | 44.9 | 38.2 |
| | GPTQ | 49.0 | 28.2 | 53.3 | 61.3 | 40.5 | 36.2 | 37.3 | 50.9 | 44.6 | 22.8 |
| | AWQ | 48.5 | 27.8 | 52.7 | 62.3 | 40.1 | 35.6 | 35.3 | 50.4 | 44.1 | 27.1 |
| | OmniQ | 50.2 | 29.4 | 53.9 | 61.5 | 41.6 | 36.4 | 43.2 | 50.2 | 45.8 | 20.5 |
| | LLM-QAT | 44.7 | 28.7 | 53.7 | 60.6 | 41.1 | 34.6 | 34.9 | 50.2 | 43.5 | 37.5 |
| | SpinQuant | 50.9 | 30.8 | 46.7 | 62.1 | 41.5 | 37.3 | 39.1 | 48.9 | 44.7 | 17.6 |
| | ParetoQ | 53.5 | 33.7 | 56.1 | 65.6 | 41.7 | 40.0 | 41.2 | 51.3 | 47.9 | 21.6 |
| MobileLLM-350M | FP | 65.5 | 42.3 | 57.4 | 71.0 | 43.5 | 53.3 | 47.3 | 58.3 | 54.8 | 10.4 |
| | RTN | 58.8 | 35.9 | 59.5 | 65.0 | 41.8 | 43.9 | 39.1 | 53.8 | 49.7 | 37.4 |
| | GPTQ | 59.8 | 34.0 | 60.6 | 67.5 | 42.1 | 46.5 | 38.7 | 53.9 | 50.4 | 14.0 |
| | AWQ | 59.5 | 35.7 | 57.5 | 66.9 | 42.1 | 47.0 | 42.3 | 53.8 | 50.6 | 14.5 |
| | OmniQ | 58.0 | 36.2 | 61.2 | 67.2 | 42.4 | 46.1 | 42.1 | 52.0 | 50.7 | 13.5 |
| | LLM-QAT | 54.6 | 35.4 | 60.5 | 65.9 | 42.2 | 42.6 | 41.9 | 53.4 | 49.5 | 22.6 |
| | SpinQuant | 57.9 | 35.3 | 59.3 | 67.0 | 41.4 | 47.5 | 43.2 | 54.3 | 50.7 | 12.1 |
| | ParetoQ | 63.9 | 40.5 | 61.4 | 70.6 | 43.2 | 51.4 | 50.0 | 56.6 | 54.7 | 14.9 |
| MobileLLM-600M | FP | 68.5 | 47.6 | 60.5 | 72.5 | 44.4 | 59.5 | 51.4 | 61.4 | 58.2 | 9.0 |
| | RTN | 55.3 | 32.5 | 57.0 | 63.1 | 42.1 | 40.6 | 37.7 | 54.1 | 47.8 | 12.0 |
| | GPTQ | 61.4 | 38.0 | 55.7 | 68.5 | 42.5 | 51.8 | 43.2 | 56.2 | 52.2 | 11.7 |
| | AWQ | 63.6 | 39.5 | 55.6 | 70.0 | 43.1 | 53.0 | 45.0 | 58.0 | 53.5 | 12.9 |
| | OmniQ | 64.9 | 41.6 | 63.4 | 69.8 | 42.1 | 53.0 | 45.4 | 58.2 | 54.8 | 11.3 |
| | LLM-QAT | 61.8 | 38.0 | 62.1 | 68.5 | 43.6 | 48.9 | 44.2 | 54.6 | 52.7 | 19.0 |
| | SpinQuant | 63.4 | 42.9 | 60.9 | 68.7 | 42.4 | 52.0 | 44.5 | 57.4 | 54.0 | 10.5 |
| | ParetoQ | 68.2 | 47.4 | 64.2 | 73.1 | 44.2 | 58.1 | 50.2 | 62.4 | 58.5 | 13.2 |
| MobileLLM-1B | FP | 73.4 | 50.8 | 67.6 | 74.1 | 46.7 | 64.7 | 56.6 | 62.7 | 62.1 | 8.0 |
| | RTN | 59.7 | 36.6 | 58.9 | 67.2 | 40.8 | 45.0 | 44.3 | 53.4 | 50.7 | 19.1 |
| | GPTQ | 66.7 | 43.0 | 63.5 | 72.3 | 42.9 | 57.8 | 49.2 | 59.4 | 56.8 | 10.2 |
| | AWQ | 68.8 | 43.5 | 62.9 | 71.1 | 43.7 | 57.9 | 49.2 | 57.0 | 56.8 | 10.8 |
| | OmniQ | 69.5 | 44.7 | 64.8 | 72.1 | 43.5 | 57.3 | 47.0 | 57.7 | 57.1 | 9.8 |
| | LLM-QAT | 65.3 | 42.6 | 61.2 | 70.4 | 44.0 | 54.3 | 48.8 | 56.8 | 55.5 | 17.4 |
| | SpinQuant | 68.2 | 44.0 | 63.5 | 71.1 | 43.9 | 57.2 | 45.7 | 59.0 | 56.6 | 8.9 |
| | ParetoQ | 72.3 | 51.4 | 67.0 | 74.5 | 45.7 | 63.4 | 53.7 | 62.1 | 61.3 | 12.4 |
| MobileLLM-1.5B | FP | 73.9 | 51.4 | 70.0 | 74.8 | 46.6 | 66.4 | 55.1 | 63.2 | 62.7 | 7.8 |
| | RTN | 63.2 | 38.0 | 58.5 | 67.2 | 43.6 | 47.9 | 45.9 | 56.0 | 52.5 | 10.2 |
| | GPTQ | 70.6 | 43.7 | 64.5 | 71.9 | 45.0 | 59.2 | 50.8 | 58.9 | 58.1 | 9.9 |
| | AWQ | 72.6 | 46.8 | 66.0 | 71.7 | 44.6 | 61.7 | 52.0 | 62.4 | 59.7 | 9.6 |
| | OmniQ | 71.8 | 46.4 | 67.4 | 72.9 | 46.2 | 60.9 | 50.2 | 61.9 | 59.7 | 9.1 |
| | LLM-QAT | 68.6 | 44.4 | 62.4 | 71.8 | 45.4 | 57.8 | 49.2 | 57.2 | 57.1 | 15.4 |
| | SpinQuant | 71.5 | 45.1 | 67.8 | 71.9 | 44.8 | 61.3 | 50.2 | 61.6 | 59.3 | 8.5 |
| | ParetoQ | 72.6 | 49.9 | 70.6 | 75.7 | 47.7 | 66.0 | 56.2 | 64.5 | 62.9 | 11.4 |
| LLaMA-1B | FP | 64.8 | 42.5 | 64.8 | 74.8 | 44.8 | 64.4 | 50.2 | 61.5 | 58.5 | 9.6 |
| | RTN | 28.9 | 25.0 | 55.9 | 53.5 | 37.8 | 30.1 | 28.9 | 50.6 | 38.8 | 30.9 |
| | GPTQ | 37.4 | 27.3 | 43.1 | 58.4 | 39.2 | 37.1 | 32.4 | 53.8 | 41.1 | 68.6 |
| | AWQ | 41.5 | 26.7 | 49.2 | 58.0 | 41.4 | 34.9 | 31.8 | 52.8 | 42.0 | 1.5e2 |
| | OmniQ | 39.0 | 28.8 | 61.3 | 58.8 | 40.0 | 36.3 | 32.9 | 52.7 | 43.7 | 63.4 |
| | LLM-QAT | 52.7 | 32.4 | 60.5 | 66.6 | 44.0 | 43.2 | 40.2 | 53.8 | 49.2 | 20.7 |
| | SpinQuant | 56.9 | 34.9 | 61.0 | 69.3 | 42.0 | 53.4 | 41.2 | 56.2 | 51.9 | 12.6 |
| | ParetoQ | 65.3 | 41.9 | 64.2 | 73.8 | 43.9 | 61.3 | 47.7 | 59.5 | 57.2 | 10.9 |
| LLaMA-3B | FP | 72.6 | 50.7 | 74.6 | 78.2 | 48.5 | 74.3 | 53.7 | 69.2 | 65.2 | 7.7 |
| | RTN | 40.4 | 29.7 | 60.1 | 60.6 | 41.3 | 43.4 | 33.4 | 52.9 | 45.2 | 24.9 |
| | GPTQ | 50.4 | 34.6 | 65.1 | 66.6 | 44.1 | 53.8 | 35.7 | 58.8 | 51.1 | 11.4 |
| | AWQ | 58.5 | 36.5 | 65.4 | 70.8 | 43.1 | 54.8 | 44.6 | 59.3 | 54.1 | 37.7 |
| | OmniQ | 59.7 | 38.6 | 47.6 | 73.5 | 45.9 | 62.4 | 49.8 | 61.8 | 54.9 | 12.7 |
| | LLM-QAT | 64.4 | 40.1 | 62.0 | 71.7 | 45.0 | 58.2 | 44.7 | 59.9 | 55.8 | 13.4 |
| | SpinQuant | 66.4 | 43.8 | 70.8 | 73.9 | 47.7 | 67.6 | 51.0 | 67.1 | 61.0 | 9.2 |
| | ParetoQ | 72.3 | 49.8 | 73.3 | 76.7 | 48.8 | 71.9 | 56.2 | 67.3 | 64.5 | 8.4 |
| LLaMA-8B | FP | 81.0 | 57.7 | 83.6 | 81.0 | 49.3 | 79.5 | 55.7 | 73.9 | 70.2 | 6.2 |
| | RTN | 42.4 | 29.4 | 43.0 | 61.6 | 41.0 | 37.3 | 34.2 | 53.9 | 42.9 | 12.6 |
| | GPTQ | 60.8 | 35.5 | 69.0 | 70.3 | 44.9 | 61.3 | 38.7 | 64.9 | 55.7 | 9.1 |
| | AWQ | 72.3 | 46.1 | 74.9 | 75.9 | 48.2 | 70.8 | 52.0 | 66.8 | 63.4 | 16.6 |
| | OmniQ | 68.0 | 45.4 | 68.3 | 73.9 | 46.0 | 68.7 | 50.4 | 62.3 | 60.4 | 12.1 |
| | LLM-QAT | 68.8 | 48.8 | 71.1 | 75.9 | 46.8 | 67.8 | 48.2 | 65.1 | 61.6 | 10.5 |
| | SpinQuant | 75.5 | 52.0 | 81.0 | 78.7 | 49.2 | 74.3 | 53.6 | 70.5 | 66.9 | 7.4 |
| | ParetoQ | 78.2 | 55.7 | 80.6 | 80.2 | 50.1 | 76.5 | 55.1 | 70.9 | 68.4 | 7.0 |

$[0, 0.01, 0.05, 0.1, 0.2, 0.4, 0.6, 0.8, 0.9, 0.95, 0.99, 1]$. Here, a ratio of 0 corresponds to QAT from scratch, while a ratio of 1 represents full-precision training followed by post-training quantization (PTQ).

For full-precision training, we use 8×8 GPUs, a batch size of 16, a weight decay of 0.1, an initial learning rate of $2.5 \times 10^{-3}$, and a linear learning rate decay to zero. For quantized network training, we also use 8×8 GPUs but with a batch size of 8, no weight decay, an initial learning rate of $1 \times 10^{-4}$, and a linear learning rate decay to zero.

## A.6 Embedding Bit Precision vs. Accuracy Trade-off

Despite the prevalent practice of not quantizing embedding and output layers, as noted in prior works such as Frantar et al. [Frantar et al., 2022] and Ma et al. [Ma et al., 2024], our study extends the

Table 5: Complete results of **4-bit quantization** on WikiText2 and Zero-shot Common Sense Reasoning tasks.

| Model Name | Method | ARC-e (↑) | ARC-c (↑) | BoolQ (↑) | PIQA (↑) | SIQA (↑) | HellaSwag (↑) | OBQA (↑) | WinoGrande (↑) | Avg. (↑) | Wiki2 (↓) |
|---|---|---|---|---|---|---|---|---|---|---|---|
| MobileLLM-125M | FP | 56.0 | 34.5 | 56.3 | 65.5 | 42.0 | 40.1 | 42.2 | 51.3 | 48.5 | 14.9 |
| | RTN | 53.4 | 33.3 | 53.9 | 64.7 | 41.5 | 39.7 | 40.2 | 51.8 | 47.3 | 9.2 |
| | GPTQ | 53.4 | 33.5 | 54.7 | 64.4 | 42.5 | 39.2 | 43.8 | 52.2 | 48.0 | 16.1 |
| | AWQ | 54.2 | 33.5 | 56.6 | 65.0 | 41.9 | 39.5 | 41.1 | 51.2 | 47.9 | 16.0 |
| | OmniQ | 52.8 | 33.5 | 56.1 | 63.4 | 41.4 | 39.2 | 39.7 | 50.8 | 47.1 | 16.1 |
| | LLM-QAT | 54.2 | 33.4 | 52.2 | 64.7 | 42.4 | 39.0 | 42.7 | 51.7 | 47.5 | 52.1 |
| | SpinQuant | 55.2 | 33.7 | 58.1 | 65.0 | 42.5 | 39.7 | 40.6 | 49.8 | 48.1 | 15.4 |
| | ParetoQ | 55.4 | 35.2 | 54.1 | 66.2 | 41.7 | 40.8 | 44.0 | 52.1 | 48.7 | 20.4 |
| MobileLLM-350M | FP | 65.5 | 42.3 | 57.4 | 71.0 | 43.5 | 53.3 | 47.3 | 58.3 | 54.8 | 10.4 |
| | RTN | 63.6 | 39.0 | 55.2 | 70.3 | 42.8 | 51.5 | 49.8 | 58.9 | 53.9 | 7.3 |
| | GPTQ | 63.8 | 39.7 | 53.7 | 69.7 | 42.7 | 51.4 | 47.9 | 57.8 | 53.3 | 11.0 |
| | AWQ | 63.0 | 38.5 | 57.1 | 70.7 | 43.6 | 51.6 | 45.8 | 55.2 | 53.2 | 11.2 |
| | OmniQ | 63.9 | 37.4 | 56.2 | 69.8 | 42.4 | 50.9 | 46.6 | 54.2 | 52.7 | 11.1 |
| | LLM-QAT | 63.4 | 42.0 | 59.8 | 70.1 | 43.6 | 51.5 | 47.0 | 57.5 | 54.4 | 17.1 |
| | SpinQuant | 62.5 | 37.8 | 56.1 | 69.6 | 43.1 | 51.5 | 43.8 | 55.7 | 52.5 | 10.6 |
| | ParetoQ | 64.9 | 41.6 | 57.8 | 71.3 | 44.4 | 53.5 | 48.2 | 57.9 | 55.0 | 14.2 |
| MobileLLM-600M | FP | 68.5 | 47.6 | 60.5 | 72.5 | 44.4 | 59.5 | 51.4 | 61.4 | 58.2 | 9.0 |
| | RTN | 67.8 | 45.1 | 48.5 | 71.6 | 44.0 | 57.8 | 49.8 | 59.6 | 55.5 | 15.4 |
| | GPTQ | 68.5 | 47.0 | 50.2 | 72.3 | 43.8 | 57.7 | 49.6 | 58.9 | 56.0 | 9.4 |
| | AWQ | 68.8 | 45.0 | 60.5 | 72.3 | 44.0 | 58.3 | 48.2 | 59.8 | 57.1 | 9.7 |
| | OmniQ | 68.4 | 45.0 | 59.5 | 71.5 | 43.7 | 58.1 | 49.0 | 59.0 | 56.8 | 9.5 |
| | LLM-QAT | 67.2 | 47.4 | 65.2 | 71.8 | 43.8 | 57.8 | 50.6 | 59.8 | 57.9 | 11.0 |
| | SpinQuant | 69.1 | 44.7 | 64.3 | 71.5 | 43.6 | 57.4 | 49.0 | 57.1 | 57.0 | 9.2 |
| | ParetoQ | 69.3 | 48.9 | 64.8 | 73.2 | 44.2 | 59.5 | 51.2 | 62.1 | 59.2 | 13.2 |
| MobileLLM-1B | FP | 73.4 | 50.8 | 67.6 | 74.1 | 46.7 | 64.7 | 56.6 | 62.7 | 62.1 | 8.0 |
| | RTN | 73.1 | 47.7 | 63.5 | 75.0 | 45.7 | 62.8 | 56.2 | 61.2 | 60.6 | 11.2 |
| | GPTQ | 72.6 | 50.7 | 65.5 | 74.8 | 45.9 | 63.7 | 56.6 | 62.3 | 61.5 | 8.4 |
| | AWQ | 73.7 | 48.6 | 65.3 | 73.5 | 45.6 | 62.5 | 49.4 | 60.6 | 59.9 | 8.5 |
| | OmniQ | 72.5 | 49.3 | 66.0 | 74.3 | 45.0 | 62.5 | 52.2 | 62.1 | 60.5 | 8.4 |
| | LLM-QAT | 72.1 | 49.5 | 66.1 | 73.9 | 46.2 | 63.0 | 55.4 | 63.7 | 61.2 | 10.0 |
| | SpinQuant | 70.5 | 47.0 | 66.6 | 74.1 | 44.2 | 62.4 | 51.6 | 61.6 | 59.8 | 8.2 |
| | ParetoQ | 74.7 | 52.1 | 67.9 | 74.8 | 46.9 | 64.8 | 56.2 | 62.1 | 62.5 | 11.7 |
| MobileLLM-1.5B | FP | 73.9 | 51.4 | 70.0 | 74.8 | 46.6 | 66.4 | 55.1 | 63.2 | 62.7 | 7.8 |
| | RTN | 73.7 | 49.5 | 66.0 | 74.5 | 46.4 | 65.5 | 52.7 | 62.0 | 61.3 | 9.4 |
| | GPTQ | 73.9 | 49.9 | 68.9 | 73.7 | 46.6 | 64.9 | 54.5 | 62.0 | 61.8 | 8.2 |
| | AWQ | 74.9 | 49.2 | 68.1 | 73.4 | 46.3 | 65.0 | 52.2 | 63.8 | 61.6 | 8.2 |
| | OmniQ | 75.3 | 50.2 | 67.6 | 74.2 | 45.8 | 64.6 | 53.8 | 62.7 | 61.8 | 8.2 |
| | LLM-QAT | 72.3 | 49.5 | 70.1 | 73.5 | 47.1 | 64.5 | 53.2 | 63.4 | 61.7 | 13.9 |
| | SpinQuant | 73.8 | 48.9 | 68.6 | 73.9 | 45.8 | 64.8 | 52.3 | 63.9 | 61.5 | 7.9 |
| | ParetoQ | 74.4 | 51.7 | 71.8 | 75.3 | 47.3 | 67.2 | 57.6 | 63.0 | 63.6 | 11.0 |
| LLaMA-1B | FP | 64.8 | 42.5 | 64.8 | 74.8 | 44.8 | 64.4 | 50.2 | 61.5 | 58.5 | 9.6 |
| | RTN | 55.7 | 36.3 | 61.9 | 70.4 | 43.0 | 56.9 | 39.3 | 55.5 | 52.4 | 13.9 |
| | GPTQ | 55.2 | 38.8 | 57.9 | 70.5 | 43.5 | 55.4 | 43.2 | 58.0 | 52.8 | 13.4 |
| | AWQ | 63.4 | 40.0 | 63.5 | 73.4 | 44.5 | 60.5 | 45.8 | 60.3 | 56.4 | 12.2 |
| | OmniQ | 60.0 | 38.0 | 59.4 | 70.6 | 43.5 | 57.5 | 44.8 | 57.4 | 53.9 | 13.4 |
| | LLM-QAT | 61.3 | 38.1 | 62.3 | 73.0 | 44.2 | 59.0 | 41.8 | 58.7 | 54.8 | 8.6 |
| | SpinQuant | 62.2 | 40.3 | 64.1 | 72.3 | 44.0 | 61.6 | 47.9 | 59.8 | 56.5 | 10.3 |
| | ParetoQ | 67.4 | 43.4 | 64.4 | 74.8 | 44.4 | 63.5 | 50.4 | 61.4 | 58.7 | 10.3 |
| LLaMA-3B | FP | 72.6 | 50.7 | 74.6 | 78.2 | 48.5 | 74.3 | 53.7 | 69.2 | 65.2 | 7.7 |
| | RTN | 59.0 | 40.2 | 57.5 | 74.5 | 46.5 | 65.5 | 44.9 | 64.9 | 56.6 | 13.1 |
| | GPTQ | 64.7 | 46.7 | 66.5 | 75.3 | 47.0 | 64.7 | 50.0 | 66.7 | 60.2 | 11.1 |
| | AWQ | 69.9 | 47.6 | 72.9 | 77.2 | 49.9 | 72.8 | 51.4 | 67.5 | 63.6 | 8.7 |
| | OmniQ | 70.6 | 47.5 | 73.9 | 77.0 | 46.9 | 72.0 | 53.2 | 67.1 | 63.5 | 8.6 |
| | LLM-QAT | 71.8 | 48.1 | 74.6 | 76.6 | 48.1 | 71.4 | 52.3 | 67.4 | 63.8 | 8.2 |
| | SpinQuant | 70.2 | 47.9 | 73.8 | 76.4 | 47.8 | 71.9 | 54.3 | 68.0 | 63.8 | 8.0 |
| | ParetoQ | 73.8 | 50.3 | 75.4 | 77.2 | 48.5 | 73.3 | 57.0 | 67.7 | 65.4 | 8.0 |
| LLaMA-8B | FP | 81.0 | 57.7 | 83.6 | 81.0 | 49.3 | 79.5 | 55.7 | 73.9 | 70.2 | 6.2 |
| | RTN | 75.8 | 50.7 | 77.8 | 78.5 | 48.1 | 74.7 | 53.9 | 71.6 | 66.4 | 7.9 |
| | GPTQ | 77.7 | 51.9 | 80.6 | 79.4 | 50.8 | 76.7 | 51.8 | 71.6 | 67.6 | 7.0 |
| | AWQ | 78.5 | 51.8 | 81.8 | 80.7 | 49.2 | 78.3 | 54.2 | 72.6 | 68.2 | 7.0 |
| | OmniQ | 77.3 | 51.3 | 79.2 | 79.6 | 48.0 | 77.2 | 54.8 | 70.4 | 67.2 | 7.1 |
| | LLM-QAT | 77.4 | 54.0 | 82.9 | 79.1 | 49.2 | 77.6 | 54.3 | 72.0 | 68.3 | 13.4 |
| | SpinQuant | 78.8 | 56.0 | 82.5 | 79.7 | 49.5 | 78.5 | 54.6 | 71.5 | 68.9 | 6.5 |
| | ParetoQ | 78.6 | 55.6 | 80.2 | 80.4 | 51.5 | 77.8 | 55.7 | 71.8 | 69.0 | 6.8 |

scaling law analysis by examining the impact of quantizing these layers. As illustrated in Figure 11, utilizing 4-bit embeddings or matching the bit precision of embeddings to that of weights positions these configurations on the Pareto front, in contrast to employing 8-bit or 16-bit embeddings.

## A.7 Weight Bit Precision vs. Accuracy Trade-off

For the trade-off between weight-bit precision and model accuracy, we consider two configurations: 4-bit embeddings and embeddings with the same bit precision as weights. In both scenarios, lower-bit quantization, such as 1.58-bit, 2-bit, and 3-bit, consistently outperforms 4-bit quantization, as depicted in Figure 12.

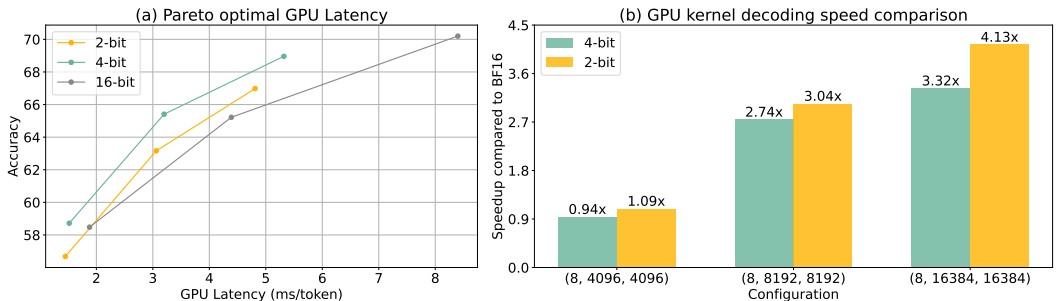

Figure 10: (a) Accuracy versus end-to-end GPU latency trade-off analysis. (b) Speedup in GPU kernel latency relative to BF16.

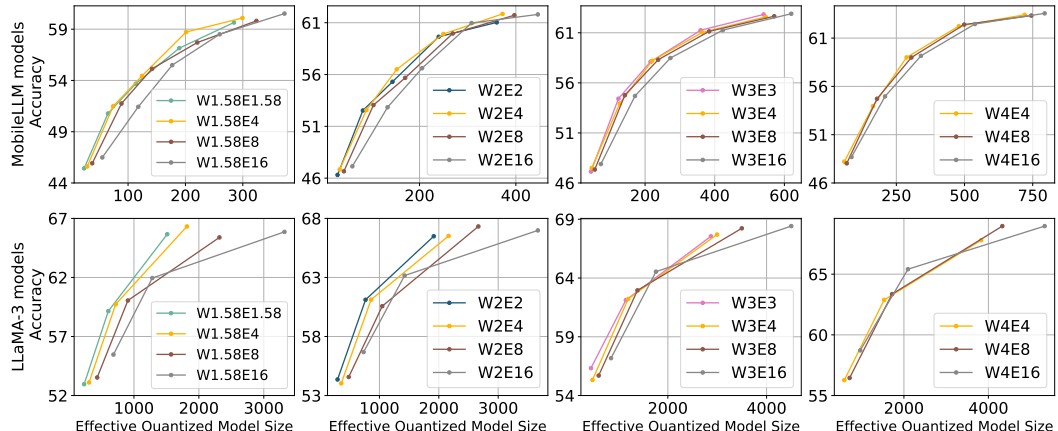

Figure 11: Trade-off between model size and accuracy for the optimal embedding bit width. "W$x$E$y$" indicates quantized weights into $x$-bits and embeddings into $y$-bits

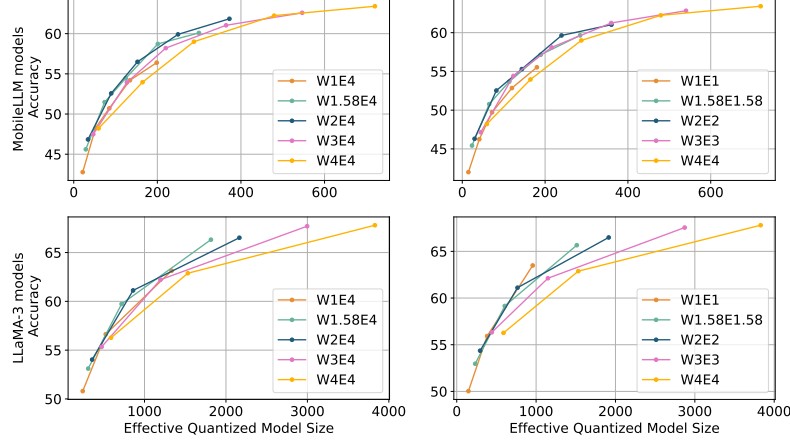

Figure 12: Trade-off between model size and accuracy for the optimal weight bit width. "W$x$E$y$" indicates quantized weights into $x$-bits and embeddings into $y$-bits

## A.8 Pareto Curve in More Tasks

Furthermore, we present results from a question-answering task, TriviaQA (TQA) [Joshi et al., 2017], and a reading comprehension benchmark, RACE [Lai et al., 2017], in Figures 13 The findings are consistent across these tasks: 1-bit quantization yields the lowest performance, whereas 1.58-bit, 2-bit, and 3-bit quantization are comparable and generally surpass the performance of 4-bit quantization.

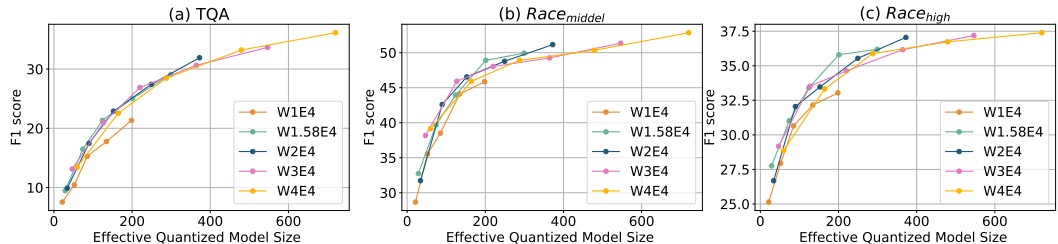

Figure 13: Trade-off between model size and accuracy on (a) TQA (b) Race$_{middle}$ and (c) Race$_{high}$. "W$x$E$y$" denotes quantized weights into $x$-bits and embeddings into $y$-bits.

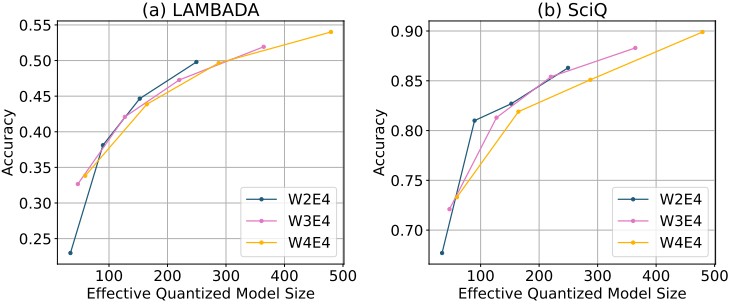

Figure 14: Trade-off between model size and accuracy on (a) LAMBADA (b) SciQ. "W$x$E$y$" indicates quantized weights into $x$-bits and embeddings into $y$-bits.

Additionally, for context-based word prediction (LAMBADA [Paperno et al., 2016]) and multiple-choice science questions (SciQ [Welbl et al., 2017]) in Figrue 14, the results also shows a clear trend of 2-bit residing on the Pareto optimal frontier, outperforming 4-bit.

### A.9 Limitations

This paper explores the Pareto frontier that characterizes the trade-off between model accuracy and size under quantization. Our findings indicate that lower-bit quantization performance is more sensitive to the fine-tuning data compared to higher-bit quantization. While our model's performance was evaluated in an academic setting, real-world applications may involve diverse training data and weight distributions. Consequently, the optimal bit-width may vary across use cases. ParetoQ provides a tool to assess a model's performance across different quantization levels. Although we identify 2-bit quantization as a promising alternative to 4-bit, the generalizability of bit-width selection should be evaluated on a case-by-case basis in practical scenarios.

### A.10 Broader Impacts

This work presents low-bit quantization for Large Language Models (LLMs). By lowering the precision required for inference, this technique can help reduce energy usage, making it a promising strategy for cutting down the computational costs of deploying LLMs.

