# OpenReview forum: "ParetoQ: Improving Scaling Laws in Extremely Low-bit LLM Quantization"
_NeurIPS.cc/2025/Conference — NeurIPS 2025 poster_

### Official Review · Reviewer_JKLc · 2025-06-28

**Clarity:** 3
**Significance:** 3
**Originality:** 3
**Rating:** 4
**Confidence:** 3

**Summary:**

This paper presents ParetoQ, a novel framework that systematically compares various quantization functions specifically tailored for extremely low-bit quantization in large language models (LLMs).

Specifically, the article provides insights and practical quantization methods for binary, ternary, and 2/3/4-bit quantization to achieve the best accuracy-efficiency tradeoff, and also provides a series of LLaMA-3 efficient quantization schemes based on ParetoQ.

Extensive experiments verify the effectiveness of the proposed quantization strategies.

**Questions:**

Please refer to weakness.

**Ethical Concerns:**

["NO or VERY MINOR ethics concerns only"]

**Final Justification:**

Thanks for the reply. I will keep my score.

**Limitations:**

Yes.

**Paper Formatting Concerns:**

No.

**Quality:**

3

**Strengths And Weaknesses:**

Strength:\
1.	The paper proposes the ParetoQ framework, which enables a systematic comparison of sub-4-bit quantization methods and provides new ideas for the design and selection of quantization functions. The paper proposes the ParetoQ framework, which enables a systematic comparison of sub-4-bit quantization methods and provides new ideas for the design and selection of quantization functions.\
2.	The paper is well-written and easy to follow.\
3.	Experiments show that ParetoQ outperforms existing quantization methods and is more efficient in hardware adaptability.\


Weakness:\
1.	As mentioned in A.9 Limitations, although the paper explores the advantages of low-bit quantization, in practical applications, the diversity of different tasks and datasets may affect the generalizability of its results.\
2.	The design of the method relies on the Pareto curve, which is the trade-off between accuracy and speed. However, if the metric for measuring the model's ability is not accuracy, but some other task metric like FID in diffusion model, is the method still applicable? If possible, it is recommended to add such a discussion.

---

> ### Author Rebuttal · Authors · 2025-07-29
>
> We thank the reviewer for  acknowledging ParetoQ is a novel framework that systematically compares various quantization functions specifically tailored for extremely low-bit quantization in LLMs. Below, we respond to the reviewers’ comments and questions in detail.
>
> **[Regarding generalizability of the results]**
>
> Thanks for the question. What we mean in A.9 Limitations is that the optimal bit may vary in real scenarios, but our method is generalizable and provides a unified framework that allows practitioners to fairly compare different bit-width choices in their own setting and identify what works best for their specific use case.
>
>
> **[Regarding extending to other task metrics beyond accuracy, such as FID in diffusion models]**
>
> Thank you for this thoughtful suggestion. We agree that extending the discussion to metrics beyond accuracy would be valuable. Indeed, we have already included generative benchmarks in the Appendix. Specifically, for the *TriviaQA* benchmark (Figure 13), we report the F1 score instead of accuracy.
>
> However, expanding the scope from LLMs to diffusion models would require solid supporting experiments, which entails non-trivial efforts (e.g., rebuilding the entire SoTA training pipeline from data preparation to fine-tuning). Without such experiments, we feel it would be irresponsible to make speculative claims. Nevertheless we will mention it as a promising direction for future work, highlighting the potential applicability of ParetoQ to other domains and metrics.
>
> *If our response has clarified your question, we’d be truly grateful if you could consider further raising your score. Thanks!*

---

### Official Review · Reviewer_hvF4 · 2025-06-29

**Clarity:** 4
**Significance:** 3
**Originality:** 2
**Rating:** 5
**Confidence:** 3

**Summary:**

The paper introduces, ParetoQ is a framework that compares different AI model quantization methods (1-4 bit) to find the best balance between size and accuracy. It shows that 2-bit and 3-bit quantization offer the optimal trade-off, outperforming previous methods.

**Questions:**

- Does the 90% observation from Figure 3 holds for bigger models?

- Regarding the observed difference in distribution for 1bit and 2 bit models. Why does finetuning help if the final distribution is different from the initial one?

**Ethical Concerns:**

["NO or VERY MINOR ethics concerns only"]

**Final Justification:**

The authors have addressed my concerns, hence I have increased my score accordingly.

**Limitations:**

Limitation section is present. The authors are encouraged to move it to the main paper.

**Paper Formatting Concerns:**

No concerns

**Quality:**

3

**Strengths And Weaknesses:**

**Strenths:**

- The paper is well written and easy to follow

- The results obtained using the proposed approach are good, comparing favorable against state-of-the-art

- Presence of large scale experiments

**Weaknesses:**

- Limited technological novelty, i.e. the quantization function and scheme, the training strategy etc., are all adopted from prior works. I do understand however that this is a study mainly and has a slightly different focus.

- Many of the findings are known or straight-forward (e.g. of known: finetuning from FP16 model, higher quantization requires longer training scheduler, sensitivity to the quantization function). Some of this have large bodies of work behind, e.g: what is the best quantization function for low-bit quantization.

- Some important, relevant works that study similar aspects were omitted, e.g:

[1] Low-Bit Quantization Favors Undertrained LLMs: Scaling Laws for Quantized LLMs with 100T Training Tokens, Xu Ouyang, Tao Ge, Thomas Hartvigsen, Zhisong Zhang, Haitao Mi, Dong Yu, 2024
[2] Continual Quantization-Aware Pre-Training: When to transition from 16-bit to 1.58-bit pre-training for BitNet language models?, Jacob Nielsen and Peter Schneider-Kamp and Lukas Galke, 2025

- The problem of optimal quantization was studies under the umbrella of mixed quantization, where the goal is to identify the best configuration with best tradeoffs. How does the approach differ in scope from this works? Some of this work also provide or aim for pareto optimal curves. The authors should seek to analyze and compare with such works. Example:

[3] One-Shot Model for Mixed-Precision Quantization, Ivan Koryakovskiy; Alexandra Yakovleva; Valentin Buchnev; Temur Isaev; Gleb Odinokikh, 2023
[4] Bit-Mixer: Mixed-precision networks with runtime bit-width selection, Adrian Bulat, Georgios Tzimiropoulos, 2024
[5] Edge Inference with Fully Differentiable Quantized Mixed Precision Neural Networks, Clemens JS Schaefer, Siddharth Joshi, Shan Li, Raul Blazquez, 2022
[6] Rethinking differentiable search for mixed-precision neural networks, Zhaowei Cai, Nuno Vasconcelos, 2020

- How does the conclusions change when considering the inference time as opposed to theoretical measures for Fig. 1?

- Unclear how general "finding 1" is. How would a different learning scheduler and hyperparameters impact this ratio? As an example, would a higher learning rate for the finetuning part change the conclusions? Does the choice of the quantizer function have any impact on this?

- Regarding "finetuning benefits across all bit-widths" where the paper suggests "pre-trained full-precision models for initialization is a more effective". This is widely known aspect for quantization in general. It's unclear why the paper presents this as a new observation, instead relevant works in the area should be cited.

---

> ### Author Rebuttal · Authors · 2025-07-29
>
> We thank the reviewer for recognizing the strength of our paper (solid experiments and outperforming previous methods) and giving us constructive suggestions on discussing more relevant works.
>
> **[Regarding the novelty]**
>
> The main novelty of our work lies in establishing a unified framework that systematically covers binary, ternary, 2-bit, 3-bit, and 4-bit quantization—each achieving state-of-the-art performance compared to prior single-bit methods. To the best of our knowledge, no previous work has provided a rigorous, apple-to-apple comparison across all these bitwidths under strictly aligned conditions, employing consistent training schemes and achieved state-of-the-art quantization functions for each bitwidth, while deriving reliable accuracy-efficiency trade-off curves. This unified framework enables a fair and comprehensive evaluation, offering insights that isolated studies focused on individual bitwidths cannot reveal. We appreciate your recognition of the distinct focus of our work and believe this framework provides unique value to the community.
>
> **[Regarding finetuning from FP16 model]**
>
> Actually, for extreme low-bit settings such as binary and ternary quantization, whether finetuning is necessary is not settled. The literature shows diverse practices and even contradictory trends. For example, in convolutional networks [7, 8], binary or ternary models are often trained from scratch, while for BERT, approaches like BinaryBERT [9] and TernaryBERT [10] usually fine-tune from a full-precision model. In contrast, recent works on LLMs, such as BitNet [11] and Spectra [12], again trained ternary LLMs directly from scratch, not for training efficiency, but for inference accuracy.
>
> This lack of consensus can be confusing, as many papers simply adopt one scheme without explicitly justifying the choice, and previous scaling law work [13] do not mention this important choice at all. To provide clearer guidance for practitioners working on extreme low-bit quantization, we believe it is worthwhile to explicitly compare this design choice under a controlled setting, as they directly affect the scaling laws and conclusions, and highlight that all bit-widths, including binary and ternary, can still benefit from finetuning.
>
> Again we are not claiming finetuning as a major discovery, instead, it builds a solid foundation for our scaling law because it achieves better accuracy than training from scratch.
>
> [7] XNOR-Net: ImageNet Classification Using Binary Convolutional Neural Networks. Mohammad Rastegari, et al.
> [8] Bi-Real Net: Enhancing the Performance of 1-bit CNNs. Zechun Liu, et al.
> [9] BinaryBERT: Pushing the Limit of BERT Quantization. Haoli Bai, et al.
> [10] TernaryBERT: Distillation-aware Ultra-low Bit BERT. Wei Zhang, et al.
> [11] BitNet: Scaling 1-bit Transformers for Large Language Models. Hongyu Wang, et al.
> [12] Spectra: Surprising Effectiveness of Pretraining Ternary Language Models at Scale. Ayush Kaushal, et al.
> [13] Scaling Laws for Precision. Tanishq Kumar, et al.
>
> **[Regarding higher quantization requires longer training scheduler]**
>
> We assume you meant *lower-bit* quantization needs a longer training schedule. We mentioned this because it is a building block in our scaling-law derivation. To keep the scaling law interpretable and practically useful, we clarify why specific training choices are necessary instead of treating them as ad hoc tricks.
>
> **[Regarding sensitivity to the quantization function]**
>
> Though important, prior scaling law studies (e.g., [1], [9]) didn’t consider quantization function sensitivity in their analysis. Other previous studies focused on single bit quantization lack cross-bit comparisons. Nevertheless, our ParetoQ achieves higher accuracy than methods specifically designed for a single bit setting.
>
> **[Some important, relevant works]**
>
> Thanks for mentioning these relevant works.
>
> [1] primarily investigates scaling laws for 2-, 3-, and 4-bit quantization, focusing on the interactions among model parameters (N), training tokens (D), and precision (P). It omits the impact of quantizer designs, which we and the reviewer both agree to be a critical aspect in low-bit quantization. And it didn’t include binary or ternary cases.
>
>  [2] investigates a training strategy for ternary pre-training specifically transitioning from 16-bit to ternary representations for BitNet-style language models. It is a study focused exclusively on ternary quantization.
>
> We will include a discussion of these works in the Related Work section and highlight the conceptual differences.
>
>
> **[How does the approach differ in scope from mixed precision quantization?]**
>
> We need to clarify that our work is not in the regime of mixed precision, and they are fundamentally different in both goal and method.
>
> Mixed-precision optimize bit allocation across layers within a fixed model using techniques like sensitivity analysis or hypernetworks. In contrast, we investigate a global scaling tradeoff: given a fixed compute or memory budget, is it better to use a smaller model with higher precision or a larger model with lower precision? Thus, we are not searching for the best bit allocation within a single architecture but analyzing how model size and uniform precision jointly affect performance.
>
> Moreover, these works [3-6] focus on ConvNets, so experimental comparison is not applicable. We will add a related work section to compare conceptual differences.
>
> **[How does the conclusions change when considering the inference time as opposed to theoretical measures for Fig. 1]**
>
> Thanks for the question. Besides accuracy-size trade-off, in Figure 7(c), we show the accuracy–latency trade-off. With properly optimized kernel implementations, 2-bit could be a promising alternative to 4-bit quantization when considering the accuracy-latency trade-off as well.
>
> **[How would different learning rates/schedulers or quantizers impact "finding 1"?]**
>
> We adopt the widely used cosine learning rate scheduler. We also compared cosine and linear schedulers, but found no significant difference in performance. We also experimented with different learning rates during the setup phase and found that the chosen configuration yields the best overall accuracy. To obtain a meaningful performance curve, we therefore used this optimal learning rate—otherwise, curves with suboptimal learning rates (and lower accuracy) would not be informative. We also observed that varying the learning rate within a reasonable range (0.5×–2×) does not significantly affect final accuracy, indicating robustness to learning rate choices.
>
> This conclusion about an effective fine-tuning ratio also holds when we switch to a different but reasonable quantization function. For example, in Figure 3, the 2-bit quantization results are obtained with SEQ; when we replace SEQ with LSQ, the same trend persists. The model achieves better performance when the FP training ratio is 90%, outperforming the 50% ratio by 0.7 points in zero-shot commonsense accuracy and exceeding both 0% and 100% by 4.1 and 10.7 points, respectively.
>
> It is important to clarify that the key takeaway is to allocate the majority of the training budget to full-precision (FP) training and approximately 10% to QAT, as we mentioned in finding 1. The 90% ratio serves as a guideline rather than a strict optimum—values like 85% or 95% could still yield comparable or even slightly better results.
>
> **[Does the 90% observation from Figure 3 holds for bigger models?]**
>
> As indicated in Figure 2, where the quantized model converges with a relatively small amount of fine-tuning tokens (<100B). For reference, the total training budget for MobileLLM is 1T tokens, and larger models like LLaMA use even more for pretraining.
>
> Therefore, the conclusion remains valid: optimal performance is achieved by dedicating the majority of the training budget to full-precision (FP) training and allocating approximately 10% to QAT.
>
> Figure 3 provides a rough range and qualitative overview, while Figure 2 presents a more detailed, model-specific, and bit-width-specific ablation. Moreover, Figure 2 includes results across different model sizes (125M, 350M, 600M, 1B, 1.5B, and 3B), supporting the generality of this trend and gives more quantitative guidelines.
>
> **[Why does finetuning help if the final distribution of 1-bit or 2-bit models differs from the initialization?]**
>
> Quantized networks, especially at very low bit-widths, operate on discretized values, which makes their loss landscapes much more rugged [14]. This increases the likelihood of getting trapped in suboptimal local minima. In contrast, full-precision networks have a smoother and more continuous loss landscape, allowing for better optimization. Fine-tuning in full precision helps position the model parameters in a more favorable region of the loss landscape before applying quantization, effectively placing the quantized model in a “better neighborhood” for further exploration.
>
> From Figure 5, we see that the distance between a full-precision model and its fine-tuned quantized counterpart ( $ ||W_{finetune} -W_{fp}||/||W_{fp}|| $ ) is around 0.4 on average for binary, ternary and 2-bit quantization, which is relatively small compared to the distance between the full-precision and its random initialization ( $ ||W_{fp} -W_{random}||/||W_{fp}|| $ ) which is $\sim 1.0$. This suggests that starting from a well-trained full-precision model provides a strong initialization that helps the quantized model reach better local minima, even if its final distribution differs from the initial one.
>
> [14] How Do Adam and Training Strategies Help BNNs Optimization? Zechun Liu, et al.
>
> *We're glad to see this work sparking so many interesting discussions, which suggests it's a promising and worthwhile direction. If our response addressed your question, we’d greatly appreciate it if you could consider further raising your scores! Thanks!*

---

> > ### Comment · Reviewer_hvF4 · 2025-08-05
> >
> > Thank you for your reply. I have increased my score accordingly.

---

### Official Review · Reviewer_m9vG · 2025-06-30

**Clarity:** 3
**Significance:** 3
**Originality:** 3
**Rating:** 5
**Confidence:** 4

**Summary:**

The paper provides a comprehensive emprical analysis of relationship of final loss or model performance with combination of model parameters, training data, quantization training regime, quantization preicison and selection of quantization function. The authors propose a quantization framework namely ParetoQ, and achieves improved quantization performance across all bit-widths. ParetoQ is presented as first unified framework for quantization by optimizing training schemes and refining quantization functions for each specific bitwidth.

**Questions:**

- Can the authors provide a deeper intuition or empirical evidence as to why "Stretched Elastic Quant" specifically works better for 1.58-bit and 2-bit compared to other adaptive quantization methods?
- Are there any new challenges or phenomena that emerge at the scales of >8B params for extremely low-bit quantization?

**Ethical Concerns:**

["NO or VERY MINOR ethics concerns only"]

**Final Justification:**

My comments have been addressed by the authors and therefore I stick to my current rating of "Accept".

**Limitations:**

Yes.

**Quality:**

3

**Strengths And Weaknesses:**

Strengths:
- The unified framework presented in this paper is unique and novel. Thorugh this, the paper addresses a signficant gap in the literature by providing comparisons across a wide range of low bit quantization settings.
- The paper shows the ParetoQ yields state of the art models at all low bitwidths, surprassing the previous literature across different precision.
- The detailed analysis of training budget and quantization functions across different bitwidths provides valuable insights that holds significant practial value.
- The emprical study provides extensive experiments spanning across 125M to 8B model parameters and various quantization strategies. The experimental evaluation is comprehensive and provides a thorough understanding on quantization trade-off in LLMs especially at low bitwidth.

Weaknesses:
- The study focuses on model scale from 125M to 8B parameter regime so its outcomes may or may not hold for larger scale LLMs. The targeted architectures are also focusing specifically on LLAMA and MobileLLM. It is unclear whether the empirical analysis holds in practise to other architectures.
- The authors introduced SEQ for low bit scenarios to keep 0s in quantization output levels. It would be good to provide a discussion on this specific and why SEQ is better than LSQ.

---

> ### Author Rebuttal · Authors · 2025-07-29
>
> We thank the reviewer for recognizing our paper’s contributions in addressing a significant gap in the literature by providing comparisons across a wide range of low bit quantization settings. We provide our point-by-point responses to the reviewers’ feedback below.
>
> **[Regarding the generalization to larger models and other model series]**
> As the model size increases, the accuracy gap between the full-precision network and the quantized network becomes smaller, as reflected in the results table of some PTQ works like OmniQuant [1]. This is because larger models have more redundancy and are generally easier to quantize.
> ParetoQ quantization framework can be easily extended to different model families and scales. To address your question, we apply ParetoQ quantization on Qwen series, including Qwen 2.5 7B and 14B models, to provide a more comprehensive analysis. ParetoQ quantization consistently outperforms previous baselines, such as RTN and GPTQ. In addition, a 2-bit Qwen 14B model achieves slightly higher accuracy than a 4-bit Qwen 7B model. We will include these results in the paper to provide further insight into the effectiveness of ParetoQ quantization.
> |  | # Bits  | ARC-e | ARC-c | BoolQ | PIQA | SIQA | HellaSwag | OBQA | WinoGrande | Avg. |
> |-|-|-|-|-|-|-|-|-|-|-|
> | Qwen-14B |  |  |  |  |  |  |  |  |  |  |
> | RTN | 2 | 25.8 | 24.1 | 38.8 | 51.6 | 36.5 | 26.0 | 25.4 | 49.0 | 34.6 |
> | GPTQ | 2 | 48.4 | 30.5 | 60.4 | 58.5 | 41.7 | 37.6 | 36.5 | 55.2 | 46.1 |
> | ParetoQ | 2 | 75.5 | 47.8 | 81.5 | 77.4 | 52.3 | 78.1 | 45.8 | 66.4 | **65.6** |
> | RTN | 4 | 75.7 | 45.5 | 78.6 | 81.1 | 48.6 | 78.6 | 38.5 | 64.8 | 63.9 |
> | GPTQ | 4 | 74.0 | 45.4 | 80.2 | 81.6 | 51.6 | 81.0 | 42.9 | 70.2 | 65.8 |
> | ParetoQ | 4 | 73.3 | 49.4 | 77.8 | 81.3 | 54.2 | 79.3 | 53.0 | 73.2 | **67.7** |
> |  |  |  |  |  |  |  |  |  |  |  |
> | Qwen-7B |  |  |  |  |  |  |  |  |  |  |
> | ParetoQ | 4 | 77.2 | 46.4 | 79.7 | 79.5 | 53.6 | 74.1 | 43.3 | 67.5 | 65.1 |
>
> [1] OmniQuant: Omnidirectionally Calibrated Quantization for Large Language Models
>
> **[Regarding more discussion on why SEQ is better than LSQ]**
> Thanks for the question. In extremely low-bit quantization, the key challenge is to maximize the representational capacity of a very limited number of quantization levels. From an information entropy perspective, a more balanced distribution of quantized values increases the entropy of the quantized output, thereby retaining more information from the original full-precision weights, statistically speaking.
>
> - In the 2-bit case:
>
> LSQ levels are {−2,−1,0,1}, and SEQ levels are {−1.5,−0.5,0.5,1.5}.
>
> In LSQ, explicitly including a zero level causes an imbalance in the distribution of quantization levels, leaving only one level on the positive side. This restricts the representation capability for positive values. It is an especially critical drawback when only 4 levels are available in 2-bit quantization.
>
> In contrast, SEQ enhances overall information capacity by using symmetric positive and negative levels with equal spacing, which encourages a more uniform level occupancy. A more uniform level distribution yields higher entropy: $H = - \sum p_i \log p_i $.  If $p_i$​ is close to uniform, H is maximized. Therefore, SEQ tends to preserve more of the signal diversity under the same bitwidth.
>
> - In the 1.58-bit case:
>
> LSQ scales weight to [-1,1], and maps [−1.0,−0.5] $\rightarrow$ -1, [−0.5,0.5] $\rightarrow$ 0, [0.5,1.0] $\rightarrow$ 1. Here, the middle bin (−0.5,0.5) has double the width of the side bins, causing more concentration on the 0 level and lowering entropy.
>
> SEQ scales weight to [-1.5, 1.5], and maps [−1.5,−0.5]→ -1, [−0.5,0.5] → 0, [0.5,1.5] → 1. All bins are equal-width, encouraging a more even probability distribution across levels, which maximizes entropy and better utilizes the limited quantization budget.
>
> While both LSQ and SEQ learn the scaling factor and the range changes dynamically during training, the initial quantization grid matters because it defines the early-stage optimization landscape. SEQ starts with a more entropy-preserving quantization grid, making it easier to retain information and stabilize training in extremely low-bit scenarios.
>
> - In short:
>
> LSQ favors noise suppression by keeping 0 but sacrifices entropy.
>
> SEQ favors balanced level usage, higher entropy, and better information retention, which is especially beneficial for ultra-low-bit quantization.
>
> *If our response addressed your question, we would greatly appreciate it if you could consider raising your score even further!*

---

### Official Review · Reviewer_RkaM · 2025-07-03

**Clarity:** 3
**Significance:** 3
**Originality:** 2
**Rating:** 4
**Confidence:** 4

**Summary:**

ParetoQ presents a unified framework for quantizing at sub-8-bit widths. They analyze the performance of quantized models as a function of dataset, number of parameters, quantization precision, quantization function, and training scheme. They present the right amount of token split for FP training, followed by QAT for the best results across different bitwidths. They also analyze different range clipping methods and show that learning based scale selection outperforms statistics-based methods across bitwidths. They propose Stretched Elastic Quant that balances output quantization levels and delivers superior performance for ternary and two-bit quantization. But by combining different quantization methods for different bitwidths, their framework is able to achieve the best-performing models at all the bitwidths under consideration.

**Questions:**

1. Are these observations specific to LLMs, or do they generalize to multimodal models?
2. What is the performance with generative tasks?
3. Why is there a drop in performance when switching from MobileLLM to Llama in Figure 2

**Ethical Concerns:**

["NO or VERY MINOR ethics concerns only"]

**Final Justification:**

The rebuttal from the authors clears most of my questions. I am keeping my favorable score.

**Limitations:**

yes

**Quality:**

4

**Strengths And Weaknesses:**

Strengths:
1. Their analysis of the performance of quantize models covers a wide range of bitwidths, model sizes, and quantization functions
2. The results from the FP training and QAT split are extensive and are useful to the broader LLM community.
3. Their framework is able to achieve the best performance across all the bitwidths and model sizes under consideration.
4. They show practical benefits of quantization with efficient kernels to achieve speed-up gains.

Weaknesses:
1. They are able to achieve the best performance by bootstrapping the best methods at different bitwidths; there is no single method that works across different bitwidths
2. The experiments are focused on discriminative or general language understanding tasks; generative tasks are not included
3. All the observations are based on empirical results, questioning the generalizability of their framework. For example, based on these methods, what can we say about larger models beyond 8B?
4. Do the same observations hold for Vision models or multimodal models?

---

> ### Author Rebuttal · Authors · 2025-07-29
>
> We thank the reviewer for the overall positive feedback on our paper, and especially for recognizing our main contributions and novelty. We address the reviewer’s comments and questions as follows:
>
> **[W1: No single method that works across different bitwidths.]**
> This observation actually reflects an inherent property of ultra-low-bit quantization, not a weakness of our method. Other literature [1] also mentioned that sub-4-bit quantization regimes are qualitatively different due to their drastically constrained representational capacity. For instance, binary quantization fundamentally collapses the value space to two levels, which makes its optimization dynamics fundamentally distinct from 2 - 4 bits. Likewise, from 2-bit to 3-bit quantization, it also exhibits different trade-offs between dynamic range and quantization error due to the significant increase in the number of quantization levels from 4 levels to 8 levels.
> ParetoQ adopts a unified core framework, utilizing a learnable scaling factor, while explicitly tailoring the quantization grid for each bitwidth because it is provably necessary.
> [1] QuEST: Stable Training of LLMs with 1-Bit Weights and Activations.
>
> **[W2 & Q2: Generative tasks.]**
> Thanks for the question. In fact, we have already included generative benchmarks in the Appendix. Specifically, TriviaQA (Figure 13) and LAMBADA (Figure 14) are representative generative benchmarks.
>
> **[W3: Generalizability to larger models beyond 8B.]**
> As the model size increases, the accuracy gap between the full-precision network and the quantized network becomes smaller, as reflected in the results table of some PTQ works like OmniQuant [2]. This is because larger models have more redundancy and are generally easier to quantize.
> ParetoQ quantization framework can be easily extended to different model families and scales. To address your question, we apply ParetoQ quantization on Qwen series, including Qwen-2.5 7B and 14B, to provide a more comprehensive analysis. ParetoQ quantization consistently outperforms previous baselines, such as RTN and GPTQ. In addition, a 2-bit Qwen 14B model achieves slightly higher accuracy than a 4-bit Qwen 7B model. We will include these results in the paper to provide further insight into the effectiveness of ParetoQ quantization.
> |  | # Bits  | ARC-e | ARC-c | BoolQ | PIQA | SIQA | HellaSwag | OBQA | WinoGrande | Avg. |
> |-|-|-|-|-|-|-|-|-|-|-|
> | Qwen-14B |  |  |  |  |  |  |  |  |  |  |
> | RTN | 2 | 25.8 | 24.1 | 38.8 | 51.6 | 36.5 | 26.0 | 25.4 | 49.0 | 34.6 |
> | GPTQ | 2 | 48.4 | 30.5 | 60.4 | 58.5 | 41.7 | 37.6 | 36.5 | 55.2 | 46.1 |
> | ParetoQ | 2 | 75.5 | 47.8 | 81.5 | 77.4 | 52.3 | 78.1 | 45.8 | 66.4 | **65.6** |
> | RTN | 4 | 75.7 | 45.5 | 78.6 | 81.1 | 48.6 | 78.6 | 38.5 | 64.8 | 63.9 |
> | GPTQ | 4 | 74.0 | 45.4 | 80.2 | 81.6 | 51.6 | 81.0 | 42.9 | 70.2 | 65.8 |
> | ParetoQ | 4 | 73.3 | 49.4 | 77.8 | 81.3 | 54.2 | 79.3 | 53.0 | 73.2 | **67.7** |
> |  |  |  |  |  |  |  |  |  |  |  |
> | Qwen-7B |  |  |  |  |  |  |  |  |  |  |
> | ParetoQ | 4 | 77.2 | 46.4 | 79.7 | 79.5 | 53.6 | 74.1 | 43.3 | 67.5 | 65.1 |
>
> [2] OmniQuant: Omnidirectionally Calibrated Quantization for Large Language Models
>
> **[W4 & Q1: Do the same observations hold for Vision models or multimodal models?]**
> Thanks for the question. Our current study focuses on language models, and we have not yet extended QAT experiments to vision or multimodal models. We refrain from making conclusions or even strong hypotheses without empirical evidence, because VLMs differ significantly from LLMs in both architecture and data distributions.
> Extending QAT to VLMs is not a straightforward adaptation of the LLM pipeline. It would require reconstructing a full SoTA VLM or multimodal training pipeline, including procedures like large-scale image-text pretraining and multimodal finetuning.
> Therefore, analyzing Pareto-style trade-offs for VLMs would warrant a dedicated separate study. While this is out of scope for the current paper, we view it as an important and promising direction for future work.
>
> **[Q3: Why is there a drop in performance when switching from MobileLLM to Llama in Figure 2?]**
> That is because the baseline full-precision MobileLLM‑1B achieves higher accuracy than LLaMA‑1B on the evaluated benchmarks. MobileLLM is specifically optimized for constrained parameter size scenarios, leading to better accuracy at the same parameter scale. In contrast, LLaMA‑1B inherits a scaled-down version of the original LLaMA design, which is not fully optimized for the 1B parameter regime.
>
> *If our response has addressed your concerns, we would greatly appreciate it if you would consider further raising your scores. Thanks!*

---

> > ### Comment · Reviewer_RkaM · 2025-08-08
> > **Keeping my score**
> >
> > Thank you for the rebuttal from the authors, which clears most of my questions. I am keeping my favorable score.

---

### Decision · Program_Chairs · 2025-09-17

**Decision:**

Accept (poster)

**Comment:**

Tis paper presents ParetoQ, a unified scaling law framework that considers a flexible schedule jointly leveraging full-precision pre-training and quantization-aware training (QAT) to train and quantize large language models (LLMs) with different bit-width settings (such as binary, ternary, and 2/3/4-bit quantization). By the proposed method with refining quantization functions, the authors claim that the best accuracy and training-efficiency tradeoff could be achieved.

The paper was initially scored (4,5,4,4) by four reviewers, who are in agreement that the core contribution of this work is technical sound. Their initial concerns are about: 1) all the observations are based on empirical results, limited to tested LLMs (small Llama and MobileLLM models only); 2) the study focuses on a relatively narrow model size ranging from 125M to 8B parameters; 3) many of the observations are known or straight-forward; 4) technical novelty is not strong.

The authors provided detailed responses to these concerns, which were well recognized by all of four reviewers. Finally, all reviewers maintained their positive scores, and the paper got the scores of (4,5,5,4). The AC read the paper, the reviews, the rebuttal, the author-reviewer discussion and the reviewers' feedback, and generally agree with the reviewers' assessment. As the proposed method cannot be easily adopted in practice due to its requirement of huge computational resources (compared to low-cost post-training quantization (PTQ) methods for LLMs with different model architectures, this work needs full-precision pre-training and QAT fine-tuning together to attain quantization goal and only limited LLM structures with small model size are studied in experiments), I recommend to accept this paper as a poster. The authors are encouraged to carefully consider the reviewers' comments/suggestions and their rebuttal in the final paper revision.